

# Simulating vertical phytoplankton dynamics in a stratified ocean using a two-layered ecosystem model

Qi Zheng [1], Johannes J. Viljoen [1], Xuerong Sun [1], and Robert J.W. Brewin [1]

[1]Centre for Geography and Environmental Science, Department of Earth and Environmental Sciences, Faculty of Environment, Science and Economy, University of Exeter, Cornwall, UK

**Correspondence:** Qi Zheng (q.zheng2@exeter.ac.uk)

**Abstract.** Phytoplankton account for around half of planetary primary production and are instrumental in regulating ocean biogeochemical cycles. Around 70% of our ocean is characterised by either seasonal or permanent stratification. In such regions, it has been postulated that two distinct planktonic ecosystems exist, one that occupies the nutrient-limited surface mixed layer, and the other that resides below the mixed-layer in a low-light, nutrient-rich environment. Owing to challenges observing the planktonic ecosystem below the mixed layer, less is known about it. Consequently, it is rarely characterised explicitly in marine ecosystem models. Here, we develop a simple, two-layered box model comprising of an ecosystem (Nutrient, Phytoplankton and Zooplankton, NPZ) in the surface mixed layer and a separate one (NPZ) in a subsurface layer below it. The two ecosystems are linked only by dynamic advection of nutrients between layers and controls on light attenuation. The model is forced with surface light (modelled from top-of-atmosphere) and observations of mixed layer depth. We run our model at the Bermuda Atlantic Time-series Study site (BATS) and compare results with a 30+ year time-series of phytoplankton and nutrient observations. When compared with observations, the model simulates contrasting seasonal and interannual variability in phytoplankton in the two layers, reproducing trends post 2011 caused by ocean warming and explaining the drivers. Results lend support to the hypothesis that the euphotic zone of stratified systems can be described using two vertically separated planktonic ecosystems. Nevertheless, simulating the ecosystem in the subsurface layer was more challenging than the ecosystem in the surface mixed-layer, suggesting more work is needed to study controls on subsurface planktonic communities.

## 1 Introduction

Phytoplankton are photosynthetic single-celled microorganisms, that form the base of the oceanic food web. Their contribution to Earth's primary production is similar to terrestrial plants, accounting for approximately half of it (Longhurst et al., 1995; Field et al., 1998). These tiny organisms play a crucial role in regulating the global carbon, nitrogen and phosphorus cycles, making them vital for Earth's climate regulation (Falkowski et al., 1998; Falkowski, 2012). Over the past century, a warming climate has been reported (Allen et al., 2018), and direct (e.g. change in carbon-to-chlorophyll ratio; C:Chl ratio) and indirect (e.g. changes in stratification) impacts on phytoplankton dynamics have been identified as a consequence (e.g., Winder and Sommer, 2012; Behrenfeld et al., 2016).





Around 70 % of the ocean is characterised by either seasonal or permanent stratification, with this percentage thought to be increasing with climate change (e.g., Polovina et al., 2008; Gruber, 2011; Leonelli et al., 2022). Our understanding of phytoplankton dynamics is primarily based on satellite observations of chlorophyll-a. However, stratified oceans often feature a relatively shallow nutrient depleted mixed layer with a deep chlorophyll maximum (DCM) below it, at depths of 80–120 m where nutrient concentrations are higher (Cullen, 1982; Fasham et al., 1985), which is hidden from the satellites (Cullen, 2015; Cornec et al., 2021; Stoer and Fennel, 2024). The community of phytoplankton at the DCM is thought to contribute significantly to biogeochemical cycling in stratified waters, but remains understudied (Dai et al., 2023; Viljoen et al., 2024; Stoer and Fennel, 2024). Considering the potential expansion of stratified waters with climate change (Li et al., 2020), it is important we learn more about phytoplankton dynamics below the mixed layer.

In a stratified ocean, the euphotic zone can be divided theoretically into two layers (Dugdale, 1967). The upper zone extends from the surface to the bottom of the mixed layer, and is characterized by high light but depleted nutrients. The lower zone between the bottom of mixed layer and the euphotic zone is low in light but replete in nutrients (Eppley et al., 1973; Small et al., 1987). To understand the vertical distribution of phytoplankton in stratified waters from observations collected at sea, empirical methods have been employed to partition profiles of total phytoplankton biomass into vertically separated layers. For example, Lange et al. (2018) show that high-light adapted and low-light adapted *Prochlorococcus* dominate the surface and subsurface layers respectively in the tropical Atlantic. Brewin et al. (2022) developed an algorithm to partition phytoplankton vertically into two communities within the upper ocean of the northern Red Sea. Recently, Viljoen et al. (2024) used this algorithm to partition phytoplankton into two vertically separated communities within the Sargasso Sea (at the Bermuda Atlantic Time-series Study site (BATS)). They found that two communities exhibit distinct and contrasting responses to climate variability over multidecadal timescales. From 2011 to 2022, chlorophyll in the surface mixed layer showed a decreasing trend while chlorophyll below the mixed layer and above the euphotic zone displayed an increasing trend (Viljoen et al., 2024). Understanding the mechanisms controlling the different trends in these two vertically separated phytoplankton communities may help improve predictions of future changes to the base of the marine ecosystem in stratified waters. Exploring these mechanisms requires the development of a suitable ecosystem model.

A wide range of ecosystem models are available to the biological oceanographic community, ranging from simple single Nutrient-Phytoplankton-Zooplankton (NPZ) models to complex models with multiple nutrient, detrital and plankton state variables. Although simple, NPZ models have proven remarkably useful for both theoretical investigation of marine ecosystems and for interpretation of observations (see Franks, 2002). A NPZ box model with physical forcing (mixed layer forcing) developed by Evans and Parslow (1985) has provided useful insights into phytoplankton annual cycles within the mixed layer (see Miller and Wheeler, 2012). Building on this early work, the first nitrogen-based NPZD model (where the D refers to a detrital state variable) was developed by Fasham et al. (1990), who also proposed the most widely-used set of parameters for NPZ modelling. This model was later validated at BATS (Fasham, 1993), providing a foundation for further NPZD model development. Efforts have been devoted to further NPZD model development to address various scientific questions at BATS. For instance, Hurtt and Armstrong (1996, 1999) developed a model based on Fasham et al. (1990) to improve the simulation of chlorophyll concentrations and primary production dynamics, through comparisons with observations. Doney et al. (1996)





developed a 1D, physically coupled NPZD model, following the rationale of Fasham et al. (1990), to investigate the seasonal
interaction between physics and biology in the upper ocean at BATS. Spitz et al. (1998) assimilated observations from BATS
to refine poorly known parameters in the NPZD model of Fasham et al. (1990). Similarly, Schartau and Oschlies (2003)
assimilated observations to derive a set of parameters that enhance the NPZD model performance in different locations in the
North Atlantic Ocean, including at BATS. The NPZD model of Fasham et al. (1990) has also been coupled with a 3-D physical
modelling framework (Sarmiento et al., 1993), to simulate ecosystems at a global scale (e.g. Oschlies and Garçon, 1999; Fennel
et al., 2006; Gruber et al., 2006; Druon et al., 2010).

In this paper, we build on the early work of Dugdale (1967) and the NPZ box model of Evans and Parslow (1985), to
construct a simple two-layered vertically structured NPZ model that partitions the euphotic zone of stratified waters into a
surface high-light nutrient-depleted layer, and a subsurface low-light nutrient-replete layer, using the mixed layer depth and
base of the euphotic zone as our boundaries. Our model is built on the assumption that these two layers host two different
ecosystems (Dai et al., 2023), and are linked by dynamic advection of nutrients between layers and the attenuation of light.
We use data collected at the Bermuda Atlantic Time-series Study site (BATS, a seasonally stratified ocean) to evaluate the
construction of our model, and to test whether it can capture seasonal and multi-decadal variability in phytoplankton dynamics
within and below the mixed layer, and perhaps explain the opposite trends in these two vertically separated layers over the
2011 to 2022 period observed by Viljoen et al. (2024). A key novelty of our model is the implementation of a different set
of parameters for each layer, setting it apart from existing ecosystem models. A detailed description of the model functions,
parameters, and datasets used is provided in Section 2. The findings from sensitivity tests, model outputs, comparisons with
observations and the mechanism of different trends post 2011 in two communities are presented in Section 3. Finally, Section
4 offers a concluding discussion of our findings.

## 2 Method

### 2.1 Model description

We present a two-layered NPZ model, building on the single-layer model of Evans and Parslow (1985). Figure 1 shows a
schematic outline of our model, displaying the interaction of different nutrients, phytoplankton and zooplankton pools. The
surface layer extends from the sea surface down to the mixed layer depth ($z_m$), while the subsurface layer spans from the
bottom of $z_m$ to the euphotic depth ($z_{eu}$) representing the depth limit of the euphotic zone (EU). In this model, $z_{eu}$ is defined
as the depth at which 0.001 % of surface light level is available (note this is considerably deeper than the common definition of
the 1 % of surface light level, owing to the fact growth by phytoplankton has been observed well below the 1 % light level (Cox
et al., 2023) and to ensure that the bottom boundary of the model is always deeper than $z_m$. A key assumption is made in the
structure of our model: the two different phytoplankton and zooplankton communities, in the two layers (Dai et al., 2023), do
not interact directly. Essentially, we make the assumption that each ecosystem is adapted to the environment it resides in, and
thus takes a competitive advantage. The two ecosystems are linked only through the exchange in nutrients between two layers
as indicated by the arrows in Figure 1. Our model is run using two forcings, $z_m$ which comes from observational data at the



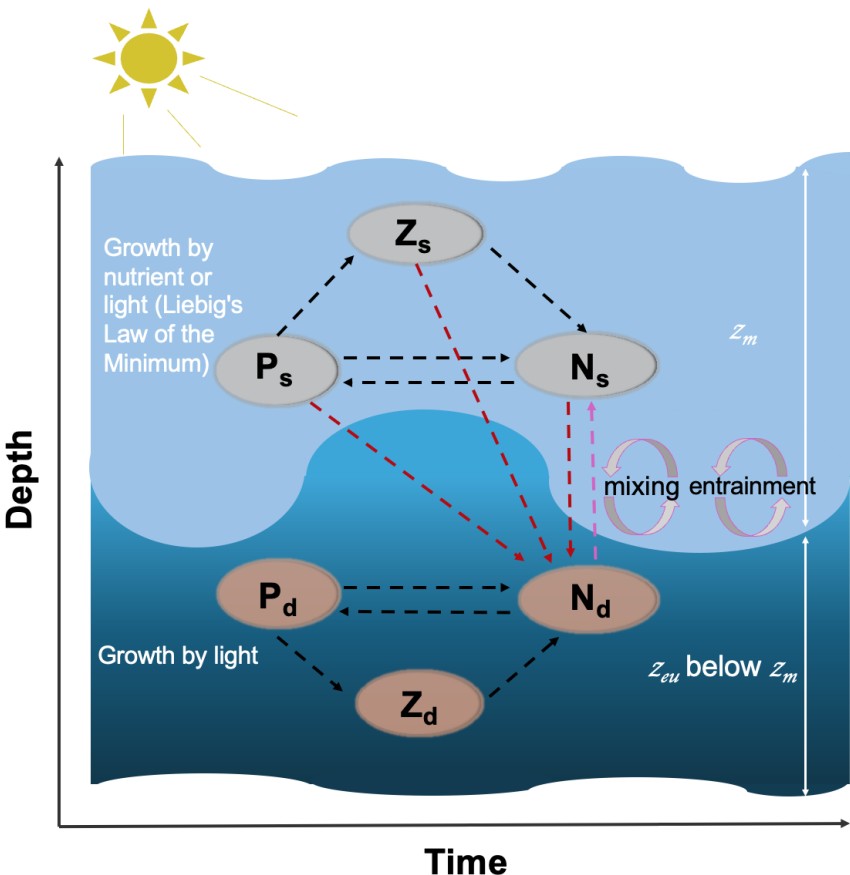

**Figure 1.** Schematic diagram of the two-layer ecosystem model. Light and dark blue shades represent the surface and subsurface layers respectively. $P_s$, $Z_s$ and $N_s$ ($P_d$, $Z_d$ and $N_d$) refer to the phytoplankton, zooplankton and nutrient pools at the surface (subsurface) respectively. $z_m$ and $z_{eu}$ refer to mixed layer depth and euphotic zone respectively. Mixing and entrainment represent the mixing and entrainment effects driven by the $z_m$, which is highlighted by the pink dashed line. Dark red dashed lines represent the nutrient excursion from the surface to the subsurface layer. The arrow direction denotes the direction of the nitrogen cycle.

Bermuda Atlantic Time-series Study site (BATS) (see description below) and broadband surface light. Following Miller and Wheeler (2012) and Brock (1981), we model daily averaged solar radiation at BATS site ($31°$) assuming clear sky conditions modulated by the atmospheric attenuation coefficient ($Atm$) and PAR fraction ($f_{par}$), and this light is attenuated through the mixed layer (with the attenuation coefficient a function of pure water and surface chlorophyll concentration) and averaged within the mixed layer, for use in surface layer modelling. Similarly, we continue to attenuate light (with the attenuation coefficient a function of pure water and subsurface chlorophyll concentration) between $z_m$ and $z_{eu}$, averaging it to represent the subsurface light forcing.





In the surface layer of our model, the rationale is similar to the model of Evans and Parslow (1985). The change in zooplankton concentration is controlled by the grazing, mortality and also the dilution and concentration effect due to fluctuations in $z_m$ (See Eq. (7)). A key modification we introduced is changing from a linear response of the zooplankton mortality term to a quadratic response, to enhance the stability of the model (e.g., Denman, 2003; Edwards and Yool, 2000; Steele and Henderson, 1992). As in Evans and Parslow (1985), the dynamics of phytoplankton concentration are controlled by phytoplankton growth and mortality, zooplankton grazing as well as the effect of $z_m$ (See Eq. (6)). For the growth term, we calculate the growth rates of nutrients and light respectively and apply Liebig's Law of the Minimum to regulate phytoplankton growth (Eq. (4)). Additionally, to ensure a conservative model, we modify the Evans and Parslow (1985) model by changing the asymmetric effect of $z_m$ to a symmetric effect to ensure a conservative model. In other words, in our model, the mixed layer incorporates both dilution and concentration effects to both nutrient and phytoplankton concentrations (See Eq. (5–6)). Furthermore, as in the Evans and Parslow (1985) model, the nutrient cycle in our model is controlled by the growth and death of phytoplankton and zooplankton, as well as the mixing and entrainment effect driven by $z_m$ (Eq. (5)). For the mixing effect, we adopt a fraction of $z_m$ ($\mu_m$ in Table 1) to represent the dynamic mixing processes between two layers following the approach of Miller and Wheeler (2012), rather than using a fixed coefficient as in the Evans and Parslow (1985) model. Considering previously used fixed coefficients, such as $0.1\,\mathrm{m\,d^{-1}}$ (Fasham et al., 1990) and $0.5\,\mathrm{m\,d^{-1}}$ (Macías et al., 2007), we select $\mu_m = 0.0055$ to yield a time-mean $\mu_m \cdot z_m$ of $0.3\,\mathrm{m\,d^{-1}}$, a mid-range value that aligns well with the $0.25\,\mathrm{m\,d^{-1}}$ used in Fennel et al. (2001).

In the subsurface layer, the dynamics of phytoplankton and zooplankton concentration adhere to principles similar to those in the surface layer. However, phytoplankton growth in the subsurface layer is simplified, and dependent solely on light (Eq. (16)), as light availability is significantly weaker and nutrients are substantially higher than the surface layer. The nutrient cycle in the subsurface layer (Eq. (17)), however, is more complex. It not only involves the phytoplankton and zooplankton cycle in the subsurface layer but also involves the phytoplankton and zooplankton excursion from the surface layer (Eq. (14)). In addition, nutrients in the subsurface layer are injected into the surface layer via mixing and entrainment processes. This exchange between nutrient pools in the surface and subsurface layers serves as the crucial link connecting the processes between the two layers.

In summary, the model equations for dissolved nutrients (N), phytoplankton (P) and zooplankton (Z) concentration dynamics in the surface and subsurface layers are shown as follows:



Surface layer:

$$\frac{dz_m}{dt} = \zeta(t), \tag{1}$$

$$\zeta^+(t) = max(0, \zeta(t)), \tag{2}$$

$$\Phi_s = \frac{a_s \epsilon_s P_{c,s}^2}{a_s + \epsilon_s P_{c,s}^2}, \tag{3}$$

$$G_s = \min\{V_{max_s}(1 - \exp(\frac{-\alpha_s I_s}{V_{max_s}})), \frac{V_{max_s} N_{c,s}}{K_s + N_{c,s}}\} \tag{4}$$

$$\frac{dN_{c,s}}{dt} = -G_s P_{c,s} + \mu_p m_s P_{c,s} + (1 - \gamma_s - \mu_g)\Phi_s Z_{c,s} + \mu_z c_s Z_{c,s}^2 + \frac{(\mu_m z_m + \zeta^+(t))(N_{c,d} - N_{c,s})}{z_m} - \frac{\zeta(t)}{z_m} N_{c,s}, \tag{5}$$

$$\frac{dP_{c,s}}{dt} = G_s P_{c,s} - m_s P_{c,s} - \Phi_s Z_{c,s} - \frac{\zeta(t)}{z_m} P_{c,s}, \tag{6}$$

$$\frac{dZ_{c,s}}{dt} = \gamma_s \Phi_s Z_{c,s} - c_s Z_{c,s}^2 - \frac{\zeta(t)}{z_m} Z_{c,s}, \tag{7}$$

$$I_k = \frac{V_{max_s}}{\alpha_s} \tag{8}$$

$$I_* = \frac{I_s}{I_k} \tag{9}$$

$$\chi_s = \frac{I_*}{\theta_m(1 - exp(-I_*))} \tag{10}$$

$$Chla_{c,s} = P_{c,s} * (C:N) * MW_C * \frac{1}{\chi_s} \tag{11}$$

Subsurface layer:

$$z_D = z_{eu} - z_m, \tag{12}$$

$$\frac{dz_D}{dt} = \eta(t), \tag{13}$$

$$\phi = \frac{(1 - \mu_z)c_s Z_{c,s}^2 z_m}{z_D} + \frac{\mu_g \Phi_s Z_{c,s} z_m}{z_D} + \frac{(1 - \mu_p)m_s P_{c,s} z_m}{z_D}, \tag{14}$$

$$\Phi_d = \frac{a_d \epsilon_d P_{c,d}^2}{a_d + \epsilon_d P_{c,d}^2}, \tag{15}$$

$$G_d = V_{max_d}(1 - \exp(-\frac{\alpha_d I_d}{V_{max_d}})) \tag{16}$$

$$\frac{dN_{c,d}}{dt} = -G_d P_{c,d} + m_d P_{c,d} + (1 - \gamma_d)\Phi_d Z_{c,d} + c_d Z_{c,d}^2 - \frac{\eta(t)}{z_D} N_{c,d} - \frac{(\mu_m z_m + \zeta^+(t))(N_{c,d} - N_{c,s})}{z_D} + \phi, \tag{17}$$

$$\frac{dP_{c,d}}{dt} = G_d P_{c,d} - m_d P_{c,d} - \Phi_d Z_{c,d} - \frac{\eta(t)}{z_D} P_{c,d}, \tag{18}$$

$$\frac{dZ_{c,d}}{dt} = \gamma_d \Phi_d Z_{c,d} - c_d Z_{c,d}^2 - \frac{\eta(t)}{z_D} Z_{c,d}, \tag{19}$$

$$Chla_{c,d} = P_{c,d} * (C:N) * \frac{1}{\chi_d} * MW_C \tag{20}$$





Where $N_{c,s}$, $P_{c,s}$, $Z_{c,s}$ and $Chla_{c,s}$ represent the dissolved nutrients, phytoplankton, zooplankton and chlorophyll-a concentration at the surface layer. Similarly, $N_{c,d}$, $P_{c,d}$, $Z_{c,d}$ and $Chla_{c,d}$ represent the dissolved nutrients, phytoplankton, zooplankton and chlorophyll-a concentration at the subsurface layer. $\Phi_s$ and $\Phi_d$ represent the grazing at the surface and subsurface

layer respectively which share the same 'Holling Type III' response (Denman and Peña, 2002) but with different parameters at different layers. $\phi$ represents the influence of the export production from the surface layer to the subsurface layer nutrient pool. $G_s$ shows the growth rate of the phytoplankton at the surface, determined by the processes most limiting to surface light ($V_{max_s}(1-\exp(\frac{-\alpha_s I_s}{V_{max_s}}))$) or nutrients ($\frac{V_{max_s} N_{c,s}}{K_s+N_{c,s}}$) (Michaelis-Menten uptake, Franks, 2002). However, at the subsurface layer, the growth of phytoplankton ($G_d$) is set to be only limited by subsurface light ($I_d$). $z_D$ is the thickness of the subsurface layer.

To calculate the surface chlorophyll concentration ($Chla_{c,s}$) in this model, we convert phytoplankton from nitrogen units in mol to carbon units in mg by multiplying by the molecular weight of carbon ($MW_C$ in Table 1) and the Redfield ratio ($C:N$ in Table 1). Then, we convert the carbon to chlorophyll using the surface carbon-to-chlorophyll ratio ($\chi_s$) following the Eq.(8-10) originating from Geider et al. (1997) and Jackson et al. (2017), where $\theta_m$ is the maximum chlorophyll-to-carbon ratio and $I_*$ is dimensionless irradiance. According to Jackson et al. (2017), a suitable value for $\theta_m$ at the surface is 0.01

gChla gC$^{-1}$. $I_*$ is defined by Eq.(9) where $I_s$ is the surface photosynthetically active radiation and $I_k$ is defined as a ratio of maximum chlorophyll-normalized production over the chlorophyll-normalized initial slope of the photosynthesis irradiance curve (Jackson et al., 2017). Since our model does not include chlorophyll-normalized parameters, we approximate $I_k$ as $\frac{V_{max_s}}{\alpha_s}$ to calculate $\chi_s$.

To calculate the subsurface chlorophyll concentration ($Chla_{c,d}$), we followed a similar method as calculating $Chla_{c,s}$, but

used a fixed C:Chl ratio in the subsurface layer ($\chi_d$). According to Viljoen et al. (2024), $\chi_d$ tends to be stable at the subsurface and shows a lower value than the surface layer. Thus we fixed $\chi_d$ as 156, half of the modelled time-mean $\chi_s$, which also falls in a range of 1-226 observed by Viljoen et al. (2024).

## 2.2   Model parameterisation

Table 1 summarises the parameters used in our model, their meanings, values, units and supporting references. In this two-layer

NPZ model, most of the basic parameters ($f_{par}$, $K_{dw}$, $\alpha_s$, $m_s$ and $\gamma_s$) at the surface come from Fasham et al. (1990) which forms the most widely-used values for NPZ modelling. $K_{dw}$ and $K_{dp_s}$ are parameters used to calculate the total attenuation coefficients ($K_{d_s}$) at the surface layer (see Eq.(21)), such that

$$K_{d_s} = K_{dw} + K_{dp_s} * Chla_{c,s}, \tag{21}$$

where $K_{dw}$ refers to the attenuation of pure water and $K_{dp_s}$ is the chlorophyll-specific light attenuation at the surface. Similarly,

$K_{dw}$ and $K_{dp_d}$ are parameters used to calculate the total attenuation coefficients ($K_{d_d}$) at the subsurface layer (see Eq.(22)), such that

$$K_{d_d} = K_{dw} + K_{dp_d} * Chla_{c,d}, \tag{22}$$

where $K_{dp_d}$ is the chlorophyll-specific light attenuation at the subsurface. In this model, we use different values for $K_{dp_s}$ and $K_{dp_d}$ as field studies have shown the chlorophyll-specific attenuation of phytoplankton changes vertically in stratified



**Table 1.** Parameters used in our two-layered NPZ model, their meanings, values, units and supporting references.

| Parameter | Symbol | Value | Unit | Reference |
|---|---|---|---|---|
| Solar constant | $SolarK$ | 1373 | $\mathrm{Wm}^{-2}$ | Miller and Wheeler (2012) |
| Atmospheric attenuation | $Atm$ | 0.5 | — | Miller and Wheeler (2012) |
| PAR fraction | $f_{par}$ | 0.41 | — | Fasham et al. (1990) |
| Light attenuation due to water | $K_{dw}$ | 0.04 | $\mathrm{m}^{-1}$ | Fasham et al. (1990) |
| Surface chlorophyll-specific light attenuation coefficient | $K_{dp_s}$ | 0.028 | $\mathrm{m}^2(\mathrm{mgChla})^{-1}$ | Uitz et al. (2008) |
| Initial value for surface nutrient concentration | $N_o$ | 0.1 | $\mathrm{mmolNm}^{-3}$ | Anugerahanti et al. (2020) |
| Initial value for nutrient concentration in the subsurface layer | $N_{d_o}$ | 2.5 | $\mathrm{mmolNm}^{-3}$ | Anugerahanti et al. (2020) |
| Initial value for phytoplankton concentration at the surface | $P_o$ | 0.2 | $\mathrm{mmolNm}^{-3}$ | Kantha (2004) |
| Initial value for zooplankton concentration at the surface | $Z_o$ | 0.25 | $\mathrm{mmolNm}^{-3}$ | Anugerahanti et al. (2020) |
| Initial value for chlorophyll concentration at the surface | $Chl_o$ | 0.1 | $\mathrm{mgm}^{-3}$ | Anugerahanti et al. (2020) |
| Initial value for phytoplankton concentration at the subsurface | $P_{d_o}$ | 0.1 | $\mathrm{mmolNm}^{-3}$ | Doney et al. (1996) |
| Initial value for zooplankton concentration at the subsurface | $Z_{d_o}$ | 0.05 | $\mathrm{mmolNm}^{-3}$ | Anugerahanti et al. (2020) |
| Initial value for chlorophyll concentration at the subsurface | $Chl_{d_o}$ | 0.13 | $\mathrm{mgm}^{-3}$ | Anugerahanti et al. (2020) |
| Initial value for mixed layer depth | $z_{m_o}$ | 52 | m | Time-mean $z_m$ at BATS |
| Initial value for euphotic zone | $z_{e_o}$ | 250 | m | Anugerahanti et al. (2020) |
| Half-saturated for phytoplankton nutrient uptake at surface layer | $K_s$ | 0.7 | $\mathrm{mmolNm}^{-3}$ | Hurtt and Armstrong (1999) |
| Initial slope of the P/I curve at surface layer | $\alpha_s$ | 0.025 | $\mathrm{day}^{-1}(\mathrm{Wm}^{-2})^{-1}$ | Fasham et al. (1990) |
| Phytoplankton mortality rate at surface layer | $m_s$ | 0.09 | $\mathrm{day}^{-1}$ | Fasham et al. (1990) |
| Phytoplankton maximum growth rate at surface layer | $V_{max_s}$ | 1.2 | $\mathrm{day}^{-1}$ | Schartau and Oschlies (2003) |
| Zooplankton assimilation efficiency at surface layer | $\gamma_s$ | 0.75 | — | Fasham et al. (1990) |
| Maximum grazing rate at surface layer | $a_s$ | 2 | $\mathrm{day}^{-1}$ | Oschlies and Garçon (1999) |
| Prey capture rate at surface layer | $\epsilon_s$ | 1 | $(\mathrm{mmolNm}^{-3})^{-2}\mathrm{day}^{-1}$ | Oschlies and Garçon (1999) |
| Zooplankton quadratic mortality rate | $c_s$ | 0.2 | $(\mathrm{mmolNm}^{-3})^{-1}\mathrm{day}^{-1}$ | Pasquero et al. (2005) |
| Dead zooplankton fraction immediately available as nutrient | $\mu_z$ | 0.2 | — | Pasquero et al. (2005) |
| Zooplankton grazing substance fraction sinking to the subsurface layer | $\mu_g$ | 0.2 | — | Pasquero et al. (2005) |
| Dead phytoplankton fraction immediately available as nutrient | $\mu_p$ | 0.2 | — | Pasquero et al. (2005) |
| Mixing fraction coefficient | $\mu_m$ | 0.0055 | — | Fennel et al. (2001) |
| Subsurface chlorophyll-specific light attenuation coefficient | $K_{dp_d}$ | 0.026 | $\mathrm{m}^2(\mathrm{mgChla})^{-1}$ | Uitz et al. (2008) |
| Initial slope of the P/I curve at subsurface layer | $\alpha_d$ | 0.256 | $\mathrm{day}^{-1}(\mathrm{Wm}^{-2})^{-1}$ | Schartau and Oschlies (2003) |
| Phytoplankton mortality rate at subsurface layer | $m_d$ | 0.05 | $\mathrm{day}^{-1}$ | Schartau and Oschlies (2003) |
| Phytoplankton maximum growth rate at subsurface layer | $V_{max_d}$ | 0.27 | $\mathrm{day}^{-1}$ | Schartau and Oschlies (2003) |
| Zooplankton assimilation efficiency at subsurface layer | $\gamma_d$ | 0.9 | — | Schartau and Oschlies (2003) |
| Maximum grazing rate at subsurface layer | $a_d$ | 1.575 | $\mathrm{day}^{-1}$ | Schartau and Oschlies (2003) |
| Prey capture rate at subsurface layer | $\epsilon_d$ | 1.6 | $(\mathrm{mmolNm}^{-3})^{-2}\mathrm{day}^{-1}$ | Schartau and Oschlies (2003) |
| Zooplankton quadratic mortality rate at subsurface layer | $c_d$ | 0.34 | $(\mathrm{mmolNm}^{-3})^{-1}\mathrm{day}^{-1}$ | Schartau and Oschlies (2003) |
| Maximum chlorophyll-to-carbon ratio at surface layer | $\theta_m$ | 0.01 | $\mathrm{gChlagC}^{-1}$ | Jackson et al. (2017) |
| C:N Redfield ratio for phytoplankton | $C:N$ | $\frac{106}{16}$ | $\mathrm{mmolC}(\mathrm{mmolN})^{-1}$ | Redfield (1958) |
| Molecular weight of Carbon | $MW_C$ | 12 | $\mathrm{mgC}(\mathrm{mmolC})^{-1}$ | — |
| C:Chl ratio at subsurface layer | $\chi_d$ | 156 | — | Half of the modelled time-mean C:Chl ratio at surface layer |

waters (Uitz et al., 2008). We approximate the chlorophyll-specific light attenuation from the chlorophyll-specific absorption




coefficient of phytoplankton derived by Uitz et al. (2008), making $K_{dp_s} = 0.028$ m$^2$(mg Chla )$^{-1}$ and $K_{dp_d} = 0.026$ m$^2$(mg Chla)$^{-1}$.

The initial value for surface phytoplankton ($P_o$) and chlorophyll ($Chl_o$) concentration is set to 0.2 mmolN m$^{-3}$ and 0.1 mg m$^{-3}$ based on observations at BATS reported by Kantha (2004) and Anugerahanti et al. (2020) respectively. On top of that,
$N_o$ is set to 0.1 mmolN m$^{-3}$ according to the time-mean of the in situ data on NO$_3$ over 1998–2007 at BATS (Anugerahanti et al., 2020). Observational data for zooplankton at different depths is more challenging to find. To keep consistency, we chose the $Z_o = 0.25$ mmolN m$^{-3}$, based on the experiment that produces phytoplankton and nutrient results closely matching in situ data (PPE in Figure 2 in Anugerahanti et al. (2020)). The initial value of mixed layer depth ($z_{m_o}$) is set to be 52 m, the time-mean value of $z_m$ from 1990 to the end of 2022 at the BATS location (data is described below). Given that the time-mean
chlorophyll tends to be near zero at 250 m (Anugerahanti et al., 2020), we define the initial value of the euphotic zone ($z_{e_o}$) as 250 m.

Regarding the initial values at the subsurface layer, $P_{d_o}$ is challenging to determine since few studies present phytoplankton concentrations in mmolN m$^{-3}$ at various depths. Nevertheless, based on the vertical phytoplankton concentration profile from 0 to 250 m at the BATS location simulated by the NPZD model from Doney et al. (1996), phytoplankton concentration at
50–250 m approximately ranges from 0.05 to 0.15 mmolN m$^{-3}$. Consequently, $P_{d_o}$ is given as 0.1 mmolN m$^{-3}$, in the centre of that range. Given that time-mean $Chla_{c,d}$ varies from 0.01 to 0.25 mg m$^{-3}$ between 50–250 m (Anugerahanti et al., 2020), $Chl_{d_o}$ is defined in the centre of the range at 0.13 mg m$^{-3}$. For $Z_{d_o}$ and $N_{d_o}$, we follow the same methodology used to determine initial values in the surface layer, but select values at around 200 m from Anugerahanti et al. (2020).

We employ the most traditional values for the maximum grazing rate ($a_s$) and prey capture rate ($\epsilon_s$) in the surface layer,
as outlined by Oschlies and Garçon (1999). The zooplankton quadratic mortality rate ($c_s$) and the coefficients related to the excretion of the nutrients from the first to the second layer ($\mu_z$, $\mu_g$ and $\mu_p$) are adopted from Pasquero et al. (2005), which is also used in other literature (e.g., Sandulescu et al., 2007; Yaya et al., 2021). Given that our hypothesis is tested at the BATS location, characterized by a seasonally stratified ocean, we adjusted the initial values, $V_{max_s}$ and $K_s$ to reflect the conditions specific to this area. Schartau and Oschlies (2003) show that $V_{max_s}$ has a strong seasonal cycle at BATS location ranging
from around 1 to 1.4 day$^{-1}$. Accordingly, we select the $V_{max_s}$=1.2 day$^{-1}$, an average value of this range. Given the range of 0.45-0.91 mmolN m$^{-3}$ for the half-saturation constant for phytoplankton nutrient uptake at the surface layer ($K_s$) at BATS, as reported by Hurtt and Armstrong (1999), we adopt $K_s$=0.7, a midpoint of that range.

In the subsurface layer, given that no study has employed different parameters at different layers in an NPZ model, it is challenging to adjust the parameters based on existing literature. However, as we know the light availability should be much
lower at the subsurface, which implies the initial slope of the photosynthesis irradiance curve ($\alpha_d$) and phytoplankton maximum growth rate ($V_{max_d}$) at the subsurface should be higher and lower than the surface respectively (Fasham, 1993). Schartau and Oschlies (2003) provide a set of parameters by assimilating observational data at the BATS location, including a higher $\alpha_d$ and lower $V_{max_d}$ which are used in our model. To maintain consistency, we use their parameters for the rest of subsurface parameters.





### 2.3 Model sensitivity analysis

To determine the sensitivity of model parameters, we conducted a series of experiments. First, the model was run from 1990 to 2022 with the parameters listed in Table 1 with the chlorophyll, zooplankton and nutrient concentration stocks at the surface and subsurface layers saved (default run). Next, the model is run again with each parameter (except for $SolarK$, $K_{dw}$, $C:N$ and $MW_C$ which are well-known and stable) individually increased and decreased by 10 % while keeping the remaining parameters fixed (sensitivity runs). We then calculate the time-mean values of each state variable in sensitivity runs and default run and computed the ratios by dividing the former by the latter. Eventually, these ratios construct a general range ($\Lambda$), illustrating how the mean values deviate from the default run in response to a 10 % change (increase or decrease) in each parameter in Table 1.

### 2.4 Model validation

Ship-based data from BATS (1990–2022) were used to calculate integrated stocks of chlorophyll and dissolved $NO_2+NO_3$, as well as mixed layer depth ($z_m$) used in this study, following the methodology outlined by Viljoen et al. (2024). Temperature profiles from CTD casts (Johnson et al., 2024) were used to compute $z_m$, specifically for profiles with concurrent temperature and salinity data that included measurements in the surface (<11 m). The $z_m$ was calculated using the temperature-based algorithm implemented in the holteandtalley Python package (Holte and Talley, 2009). Chlorophyll-a concentrations were extracted from high-performance liquid chromatography (HPLC) pigment data (Johnson et al., 2023), including only profiles with a minimum of six depth measurements and coincident CTD and particulate organic carbon (POC) bottle data. Dissolved inorganic nitrogen (DIN) measurements were obtained from nitrate+nitrite profiles on the BATS data server (http://bats.bios. edu/bats-data/). Only DIN profiles with at least six measurements in the upper 350 m and that aligned with selected HPLC chlorophyll-a and temperature profiles were used. All CTD, HPLC, and DIN data were restricted to profiles within a 0.5° latitude and longitude box around the BATS site.

We ran our model with the parameters in Table 1, from 1990-01-01 to 2022-12-31, in a daily timestep at BATS, to derive chlorophyll and nutrient concentration daily timeseries. To estimate the stocks of chlorophyll and nutrients in the model, we multiplied the vertical-averaged concentration in each layer by the thickness of each layer. The thickness of the surface and subsurface layer are $z_m$ and $z_D$ (see Eq. (12)) respectively. Next, to compare model output with observations, we used the observational chlorophyll and nutrients integration above $z_m$ timeseries at BATS from Viljoen et al. (2024) as referenced observational stocks at the surface layer, as we use the same $z_m$, which is also used as the bottom boundary for the surface layer in the model. For the subsurface layer stock calculation, we first integrate observational chlorophyll and nutrient concentrations at BATS from $z_m$ to the final depth ($z_k$) shallower than the euphotic zone ($z_{eu}$) where data is available and then divided by the ($z_k$ minus $z_m$) to obtain the averaged concentration between $z_m$ and $z_k$. This average concentration was used as a representative value for the entire water column from $z_m$ to $z_{eu}$, and then multiplied by the model subsurface thickness ($z_D$) to derive the subsurface phytoplankton and nutrient stocks.

In addition to the timeseries of stocks, we also calculate a time-mean climatology of surface and subsurface chlorophyll and nutrient stocks. To determine the relationship between observational and model outputs, we calculate Spearman's correlation





coefficient (Corr.) and the significance level ($p$) using only the time steps where data are available for both observations and model outputs.

## 2.5    Data deseasonalisation and trend analysis

To compare model estimates of interannual variability with observations, we first resampled the daily timeseries from the model to be monthly timeseries using the monthly median as described in Viljoen et al. (2024). To keep the same method that Viljoen et al. (2024) used, we dealt with missing values in observations using a median climatology. Finally, we decompose the surface and subsurface chlorophyll stocks in the model and observations and extract the deseasonalised timeseries, following the same method applied in Viljoen et al. (2024), using the Python function MSTL from the statsmodels.tsa.seasonal package, with a period of 12 representing the months of the year. We then calculate the Spearman's correlation coefficient (Corr.) and significance ($p$) of the deseasonalised timeseries between the model and observations.

To utilize this model to determine the drivers of the different post-2011 trends at the surface and subsurface layer (Viljoen et al., 2024), we first resample daily surface and subsurface light, surface C:Chl ratio, surface and subsurface chlorophyll, phytoplankton, zooplankton and DIN concentration from the model to a monthly timeseries and decomposed it to produce deseasonalised data following the method described above. To examine the trend of each timeseries post 2011, linear regression analysis (Python sklearn.linear_model.LinearRegression) is first applied to all extracted deseasonalised data described above (including modelling and observational timeseries) over 2011-2022. Based on this linear regression, we extract the slope ($S$) of the trend between 2011 (including 2011) and the end of 2022 and obtain the $p$-value to examine the significance of the trend. Also, according to the linear regression, we calculate the percentage change ($\Delta$) in the fitted model as the relative difference between the start and end values of the model fit between 2011 and 2022.

## 3    Results

### 3.1    Model sensitivity

The range of two values ($\Lambda$) for each parameter is shown in Figure 2. In the histogram, the larger $\Lambda$ varies from 1.0 the greater the difference between the adjusted and default model outputs. Among the eight state variables ($Chla_s$, $P_s$, $N_s$, $Z_s$, $Chla_d$, $P_d$, $N_d$, $Z_d$), $N_s$ show large ranges ($>$0.94–1.06) when varying 4 parameters ($N_{d_o}$, $z_{e_o}$, $K_s$ and $V_{max_s}$). This suggests that surface nutrient stocks are sensitive to changes in these parameters, reflecting their dependence on initial deep ocean nutrient stocks and surface phytoplankton growth.

Following $N_s$, chlorophyll ($Chla_d$), phytoplankton ($P_d$) and zooplankton ($Z_d$) stock in the subsurface layer exhibit the next highest sensitivity in this model. They show a tendency to be sensitive to changes in parameters at the subsurface layer (notably $m_d$, $V_{max_d}$ and $\epsilon_d$) and the light attenuation coefficient in the atmosphere ($Atm$ and $f_{par}$). In addition, $Chla_d$ is also sensitive to change in $\chi$. These suggest the net growth of phytoplankton and light play the most important role in modulating $Chla_d$, $P_d$ and $Z_d$, and the subsurface C:Chl ratio has a large impact on $Chla_d$. In contrast, surface chlorophyll ($Chla_s$) stock is sensitive





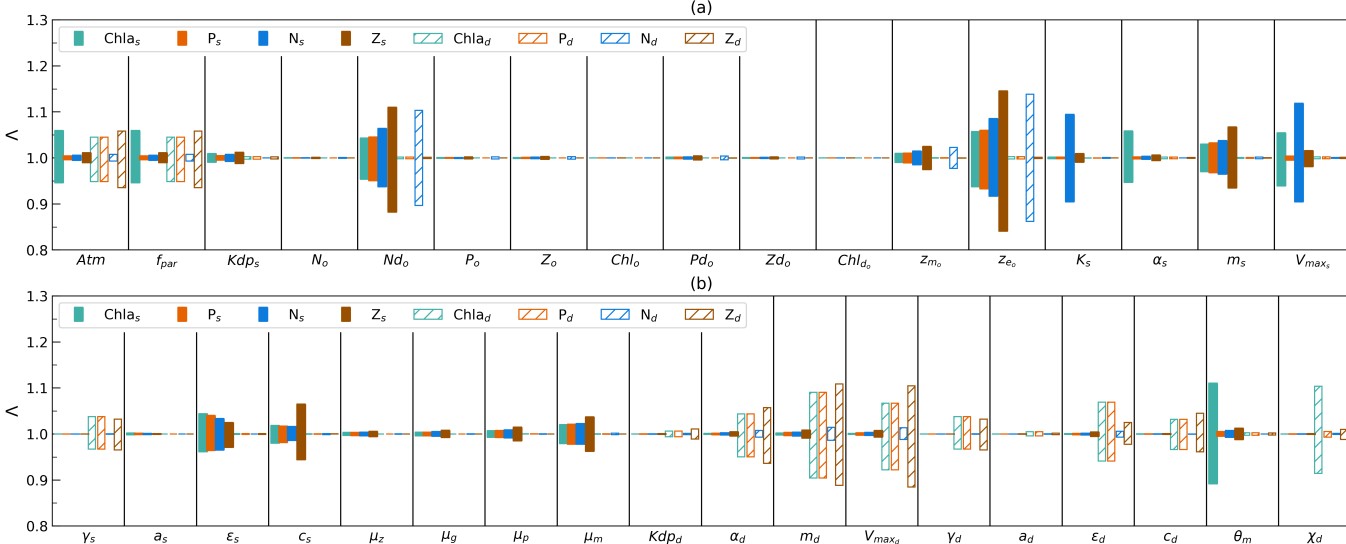

**Figure 2.** (a–b) Sensitivity of model output (difference in time-mean, denoted $\Lambda$) for surface-layer-integrated chlorophyll ($Chla_s$), phytoplankton ($P_s$), nutrients ($N_s$)and zooplankton ($Z_s$) and subsurface-layer-integrated values ($Chla_d$, $P_d$, $N_d$, $Z_d$), when increasing and decreasing each parameter from Table 1 (except for $SolarK$, $K_{dw}$, $C:N$ and $MW_C$) by 10 % individually whilst keeping the remaining values fixed.

primarily to the change in parameters at the surface layer. The key parameter $\theta_m$ in the photoacclimation model also plays an important role in determining $Chla_s$. Compared to $Chla_s$, surface zooplankton ($Z_s$) and phytoplankton stock ($P_s$) tend to be more stable, primarily sensitive to $N_{d_o}$, $z_{e_o}$ and $m_s$. This indicates that the initial assumption of the deep ocean nutrient stocks and the death of phytoplankton have a impact on $Z_s$ and $P_s$. $N_d$ tends to be the most stable parameter resilient to the changes in all parameters except for $N_{d_o}$ and $z_{e_o}$. However, notebaly, the range of $\Lambda$ in $N_d$ is the second largest (0.86–1.14) when varying $z_{e_o}$ compared to $\Lambda$ in other parameters. These findings indicate that changes in most of the parameters hardly impact $N_d$. However, the determination of $N_{d_o}$ and $z_{e_o}$ is pivotal to good $N_d$ estimation.

From a parameter perspective, this model is not sensitive to changes in the $\mu_z$, $\mu_g$, $\mu_p$ and initial values except for $z_{e_o}$ and $Nd_o$. This indicates that this model is not sensitive to changes in the excursion process from surface to subsurface layer, and changes in most of the initial values. However, $z_{e_o}$ and $Nd_o$ show large ranges of $\Lambda$ for all state variables at the surface layer (especially for surface zooplankton stock) and subsurface nutrient stock. This highlights the importance of getting initial nutrient stock conditions in the subsurface layer right.

## 3.2 Model forcing, output and validation

Figure 3 shows the daily forcing and outputs of the two-layered model from 1990 to 2022. The light forcing at the surface (solid yellow line) and subsurface layers (dashed yellow line) are presented in Figure 3a. Surface light ranges from 22 to 262 W m$^{-2}$ with the minimum in winter and the peak during summer. Subsurface light shows a similar seasonality though its



**Table 2.** The concentration of chlorophyll, phytoplankton, zooplankton and nutrients at two layers in the model. $Chla_{c,s}$, $P_{c,s}$, $Z_{c,s}$ and $N_{c,s}$ represents the chlorophyll-a, phytoplankton, zooplankton and DIN vertically averaged concentration in the surface layer timeseries, and $Chla_{c,d}$, $P_{c,d}$, $Z_{c,d}$ and $N_{c,d}$ means the chlorophyll-a, phytoplankton, zooplankton and DIN vertically averaged concentration between the bottom of the $z_m$ and $z_{eu}$ timeseries from the model. The time-mean ($\bar{R}$), minimum ($R_{min}$) and maximum ($R_{max}$) of these timeseries are presented. The units of each parameter are shown in square brackets. Given that the model stabilized in the initial month, the statistics are based on the timeseries from 1990-02-01 to 2022-12-31.

| | $Chla_{c,s}\ [mg\ m^{-3}]$ | $P_{c,s}\ [mmolN\ m^{-3}]$ | $Z_{c,s}\ [mmolN\ m^{-3}]$ | $N_{c,s}\ [mmolN\ m^{-3}]$ | $Chla_{c,d}\ [mg\ m^{-3}]$ | $P_{c,d}\ [mmolN\ m^{-3}]$ | $Z_{c,d}\ [mmolN\ m^{-3}]$ | $N_{c,d}\ [mmolN\ m^{-3}]$ |
|---|---|---|---|---|---|---|---|---|
| $\bar{R}$ | 0.07 | 0.21 | 0.18 | 0.09 | 0.06 | 0.12 | 0.10 | 1.73 |
| $R_{min}$ | 0.01 | 0.08 | 0.05 | 0.04 | 0.00 | 0.00 | 0.01 | 0.78 |
| $R_{max}$ | 0.28 | 0.62 | 1.47 | 0.50 | 0.33 | 0.65 | 0.49 | 4.91 |

intensity decreases, ranging from 0 to 11 W m$^{-2}$. The mixed layer depth ($z_m$) forcing from observations at BATS, is shown in Figure 3b (solid pink line). $z_m$ also has a strong seasonality, with a minimum in summer and a maximum in winter. It can be as shallow as 10 m during summer, and as deep as 200 m in winter. Below $z_m$, the euphotic zone ($z_{eu}$) estimated from the model is shown in Figure 3b (dashed pink line), which shows a contrasting seasonality to $z_m$ with the shallowest $z_{eu}$ in winter and deepest $z_{eu}$ in summer.

Figures 3c-f illustrate the vertically averaged concentrations of phytoplankton, zooplankton and dissolved inorganic nitrogen (DIN) within the surface layer in the model (solid lines). In the initial month of the simulation (Jan in 1990), the model exhibits instability, reflected by a spike value (0.8 mmolN m$^{-3}$) in the phytoplankton concentration at the surface layer. Nevertheless, the model quickly stabilizes. Consequently, the statistical analysis presented here is based on model outputs from 1990-02-01 to the end of 2022 (Table 2). Surface chlorophyll-a, phytoplankton, zooplankton and DIN surface concentrations show
ranges of 0.01–0.28 mg m$^{-3}$, 0.08–0.62 mmolN m$^{-3}$, 0.05–1.47 mmolN m$^{-3}$ and 0.04–0.5 mmolN m$^{-3}$ respectively, with mean values of 0.07 mg m$^{-3}$, 0.21 mmolN m$^{-3}$, 0.18 mmolN m$^{-3}$ and 0.09 mmolN m$^{-3}$ for chlorophyll, phytoplankton and zooplankton, and DIN concentration (Table 2). This model also displays seasonality in these state variables. During winter and spring, nutrient availability increases in the surface layer (with a deeper mixed layer) and slightly higher concentrations of phytoplankton and chlorophyll result. This is followed by a maximum zooplankton concentration in the subsequent month
driven by grazing on phytoplankton. These findings reveal that, at the surface layer, the phytoplankton concentration remains relatively stable over the year, but is slightly higher in spring driven by the increase in DIN and light availability, providing somewhat more food to zooplankton.

     Compared to the surface layer, subsurface chlorophyll (Figure 3c, dashed green line) concentration magnitude is very similar, reflected by a similar range of 0–0.33 mg m$^{-3}$ with a time-mean of 0.06 mg m$^{-3}$ (Table 2). However, subsurface phytoplankton
(Figure 3d, dashed orange line) and zooplankton (Figure 3e, dashed brown line) concentrations decrease to ranges of 0–0.65 mmolN m$^{-3}$ and 0.01–0.49 mmolN m$^{-3}$ respectively with the time-mean values of 0.12 mmolN m$^{-3}$ and 0.10 mmolN m$^{-3}$ respectively (Table 2). In addition, phytoplankton and zooplankton concentrations in the subsurface layer show strong seasonality, contrasting to that in the surface layer. When chlorophyll and phytoplankton concentration in the subsurface layer





**Figure 3.** (a) Solid (dashed) yellow lines refer to vertical-averaged light within the surface (subsurface) layer in the model from 1990 to 2022. (b) Solid and dashed pink lines represent the observational $z_m$ from BATS and $z_{eu}$ simulated by the model respectively from 1990 to 2022. (c) Solid (dashed) lines mean the model output of chlorophyll-a concentration vertically averaged within the surface (subsurface) layer from 1990 to 2022. (d) as in (c) but for phytoplankton concentration. (e) as in (c) but for zooplankton concentration. (f) as in (c) but for nutrient concentration.



**Table 3.** The integration of chlorophyll and nutrients from model and observation, and their relationship. $Chla_s$ and $N_s$ represent the Chl-a and DIN integration timeseries within the surface layer; $Chla_d$ and $N_d$ mean the Chl-a and DIN integration timeseries between the bottom of the $z_m$ and $z_{eu}$, $\bar{R}$, $R_{min}$, $R_{max}$ and $\sigma$ represent the time-mean, minimum, maximum, standard deviation of each timeseries described above. The correlation coefficient, $Corr.(model, obs)$, shows the relationship between the full signal of each timeseries in the model and observation; its corresponding $p$-value is shown in angle brackets. $\hat{R_{min}}$ and $\hat{R_{max}}$ means the minimum and maximum of the time-mean climatology of $Chla_s$, $Chla_d$, $N_s$ and $N_d$. $M_{min}$ and $M_{max}$ mean the month when $\hat{R_{min}}$ and $\hat{R_{max}}$ show. Values are shown for the model and observations (in parenthesis). The units of each parameter are shown in square brackets, except for $Corr.(model, obs)$ which does not have units. Given that the model stabilized in the initial month, the statistics of the full timeseries are based on the timeseries from 1990-02-01 to 2022-12-31.

| | $\bar{R}$ | $R_{min}$ | $R_{max}$ | $\sigma$ | $Corr.(model, obs)$ | $\hat{R_{min}}$ | $\hat{R_{max}}$ | $M_{min}$ | $M_{max}$ |
|---|---|---|---|---|---|---|---|---|---|
| $Chla_s$ [mg m$^{-2}$] | 5.01(5.70) | 0.14 (0.07) | 60.49(30.19) | 6.94(7.40) | 0.77 <p=0.00> | 0.52 (0.32) | 13.63(15.36) | July(July) | Jan(Jan) |
| $Chla_d$ [mg m$^{-2}$] | 17.96(26.19) | 0.00(0.00) | 107.42(94.66) | 19.69 (14.93) | 0.31 <p=0.00> | 1.89(11.57) | 39.94(35.24) | Feb(Jan) | Aug(Sep) |
| $N_s$ [mmol m$^{-2}$] | 4.71 (2.94) | 0.52(0.0) | 48.03(73.68) | 4.62(9.67) | 0.62 <p=0.00> | 1.21(0.04) | 11.04(10.25) | July (Sep) | Jan(Mar) |
| $N_d$ [mmol m$^{-2}$] | 460 (468) | 256(101) | 545(1040) | 57(157) | 0.004 <p=0.94> | 396(413) | 505(500) | Sep(Mar) | May (Nov) |

reaches the minimum, the surface concentration tends to be higher and vice versa. Minimum chlorophyll concentration in the
surface layer, coupled with the shallowest $z_m$ and highest surface light in summer, creates conditions of higher light availability
in the subsurface layer, increasing phytoplankton growth. This increased growth in subsurface phytoplankton subsequently
supports the zooplankton growth.

The DIN concentration in the subsurface layer (Figure 3f, dashed blue line) ranges from around 0.78 to 4.91 mmolN m$^{-3}$
with a time-mean value of 1.73 mmolN m$^{-3}$, which is notably higher than that in the surface layer (See Table 2). This difference
in two DIN pools and the distinctive light environments between the two layers (Figure 3a) highlight the different conditions
for phytoplankton growth: the surface layer is characterized by depleted nutrients but adequate light, whereas the subsurface
layer is dominated by weak light but abundant nutrients. These conditions likely create two distinct environments to which
different phytoplankton communities have adapted.

To compare the model output with observational data from BATS, chlorophyll and DIN stocks are calculated at each layer.
Figure 4a first shows the chlorophyll integrated from the sea surface to $z_m$ from model output (green line) and observations
(dark solid dots) spanning from 1990 to 2022. When masking the observational missing values in modelling output, the obser-
vations and modelling output show a strong and significant correlation (Corr.=0.77, Table 3). After the model stabilizes (since
1990-02-01), modelling chlorophyll ranges from 0.14 to 60.49 mg m$^{-2}$, which is higher than the observed range of 0.07–30.19
mg m$^{-2}$ as shown in Table 3. However, modelling surface chlorophyll stock shows a similar time-mean value of 5.01 mg m$^{-2}$
as in observations (5.7 mg m$^{-2}$). Also, the standard deviation from the model (6.94 mg m$^{-2}$) is very similar to in observations
(7.4 mg m$^{-2}$). These findings indicate the two-layered model successfully simulates the surface chlorophyll stock dynamics,
including the intra-annual variability.





**Figure 4.** (a) Daily chlorophyll stocks at the surface layer from the model (green solid line) and observations (dark solid dots) from 1990 to 2022. (b) Daily chlorophyll stocks at the subsurface layer from the model (green dashed line) and observations (dark hollow dots) from 1990 to 2022. (c) Daily nitrogen stocks at the surface layer from the model (blue solid line) and observations (dark hollow dots) from 1990 to 2022. (d) Daily nitrogen stocks at the surface layer from the model (blue dashed line) and observations (dark hollow dots) from 1990 to 2022. (e-h) as in (a-d) but for their time-mean climatology.

Figure 4e compares monthly averaged chlorophyll stocks above the $z_m$ from modelling output (green solid line) and observations (dark solid dots). Observed chlorophyll stocks show a strong seasonal variability with a maximum (15.36 mg m$^{-2}$) in January and a minimum (0.32 mg m$^{-2}$) in July. The model mirrors this seasonal cycle perfectly by showing a similar peak (13.63 mg m$^{-2}$) in January and a minimum (0.52 mg m$^{-2}$) in July (Table 3).



Figure 4b shows the subsurface chlorophyll integrated from $z_m$ to $z_{eu}$ from the model (green dashed line) and observations (black hollow dots). The modelling and observational timeseries also show a significant positive correlation (Corr.=0.31) although lower than the one at the surface layer. Table 3 shows that subsurface chlorophyll integration in the model ranges from
0 to 107.42 mg m$^{-2}$, closely matching the observed range of 0 to 94.66 mg m$^{-2}$. However, in general, the model estimates relatively lower chlorophyll stocks in the subsurface layer compared to observations, as indicated by a lower time-mean value of 17.96 mg m$^{-2}$ in the model than 26.19 mg m$^{-2}$ observed at BATS. Different from the mean values, the standard deviation from the model (19.69 mg m$^{-2}$) is higher than in observations (14.93 mg m$^{-2}$), which indicates the model simulates larger intra-annual variability than in the observations.

Despite simulating a similar seasonality (Figure 4f), the model shows lower stocks during spring with discrepancies in the timing of the minimum and maximum months at the subsurface layer. Table 3 shows the model predicts these extremes in February (1.89 mg m$^{-2}$) and August (39.94 mg m$^{-2}$), while observations show them in January with a higher value of 11.57 mg m$^{-2}$ and September with a similar value of 35.24 mg m$^{-2}$. The discrepancy is especially obvious from January to May. A possible explanation for the low values in the model during these months is that the averaged subsurface light is too low in the
model (Figure 3a) to simulate phytoplankton growth in the subsurface layer.

Figures 4a-b and e-f illustrate that surface and subsurface chlorophyll stocks show inverse intra-annual and seasonal variability, as seen in both the model and observations. Following a decline in surface chlorophyll seen both in the model and observations (Fig. 4e), subsurface chlorophyll stocks increase (Fig. 4f). These suggest that the distinct vertical differences in phytoplankton seasonal dynamics at BATS illustrated in observations are successfully captured by the two-layered model.

Figure 4c shows the surface DIN stocks from the model (blue solid line) and NO$_3$+NO$_2$ stocks from observations (black solid dots). After eliminating the missing values in observations in the model, the surface nitrogen stocks show a strong relationship between modelling and observational outputs, indicated by a significant correlation (Corr.=0.62, Table 3). The surface DIN stock in the model ranges from 0.52 to 48.03 mmol m$^{-2}$, smaller than the range of 0–73.68 mmol m$^{-2}$ in NO$_3$+NO$_2$ observations. This model simulates a higher time-mean DIN but smaller standard deviation (4.71±4.62 mmolN
m$^{-2}$) than in the observations (2.94±9.67 mmolN m$^{-2}$). This suggests that this model successfully simulates the surface nitrogen cycle but overestimates the time-mean nitrogen stocks and underestimates their variability.

Figure 4g shows that modelled and observational DIN stocks in the surface layer show similar seasonality. The observational NO$_3$+NO$_2$ stocks show maximum values of 10.25 mmol m$^{-2}$ in March and minimum values of 0.04 mmol m$^{-2}$ in September. The model surface DIN also shows a similar maximum of 11.04 mmol m$^{-2}$ but in January and reaches a minimum (1.21 mmol
m$^{-2}$) in July (Table 3). Figure 4g reveals that the discrepancy between model and observation is more evident in autumn and winter when the surface nutrient observations are generally extremely low, close to zero.

The subsurface DIN integration from the model (blue dashed line) and subsurface NO$_3$+NO$_2$ integration from observations (black hollow dots) are presented in Figure 4d. A small non-significant correlation coefficient (Corr.=0.004) between observational and modelling outputs suggests this model cannot simulate the variability in the NO$_3$+NO$_2$ dynamics in the subsurface
layer. This indicates the $N_d$ has large uncertainties in this model which is also reflected in the sensitivity analysis in Section 3.1. However, Table 3 shows that the model estimates a similar time-mean value of 460 mmolN m$^{-2}$ as seen in observations (468





mmolN m$^{-2}$) with a smaller standard deviation (57 mmolN m$^{-2}$) compared with observations (157 mmolN m$^{-2}$). Similarly, a smaller range in $N_d$ of 256–545 mmolN m$^{-2}$ is seen in the model compared to observations (101–1040 mmolN m$^{-2}$) as reported in Table 3. These findings suggest that this model estimates a reasonable value of DIN stocks in the subsurface layer

when compared with observations, but cannot simulate the variability in the observations.

Figure 4h also confirms the findings described above. The NO$_3$+NO$_2$ stocks in the subsurface show a minimum (413 mmolN m$^{-2}$) in March and a maximum (500 mmolN m$^{-2}$) in November in observations (Table 3). This seasonality is almost the reverse of the pattern in the surface layer seen from observations. Although this reversing pattern is not captured by the model, it estimates a similar seasonal minimum (396 mmolN m$^{-2}$) and maximum (505 mmolN m$^{-2}$).

**3.3    Drivers of post-2011 trend in chlorophyll-a in the two layers**

The deseasonalised chlorophyll stocks ($Chla_s$) at the surface layer over 1990–2022 from the model (green solid line) and observations (black solid line) are shown in Figure 5a. They show very good agreement reflected by a significant positive correlation coefficient (0.78) shown in Table 4. This coefficient even increases to 0.85 if only comparing the deseasonalised $Chla_s$ over 2011–2022 (Table 4). This suggests this two-layer NPZ model has a strong ability to simulate the interannual

variability in surface chlorophyll stocks, particularly over 2011–2022.

The observational $Chla_s$ shows a significant decreasing trend from 2011 to the end of 2022 (black straight line in Figure 5a), also reported by Viljoen et al. (2024). Table 4 confirms this significant decreasing trend by showing a slope of -0.41 mg m$^{-2}$ yr$^{-1}$ in observations. This decreasing trend post 2011 is also captured in the two-layered model (green line in Figure 5a), with a very similar trend (S=-0.42 mg m$^{-2}$ yr$^{-1}$ in Table 4) as seen in observations.

Figure 5b shows the deseasonalised subsurface chlorophyll stock timeseries ($Chla_d$) from the model (green dashed line) and observations (black dashed line). Although the correlation is weaker than seen in $Chla_s$, it is still significant (Corr. =0.44 in Table 4). This correlation becomes strong over 2011–2022, reflected by an increase in Corr. to 0.75 (Table 4). This indicates that the model is also capable of simulating interannual variability of chlorophyll in the subsurface layer, particularly post 2011, albeit weaker than at the surface layer. Looking at 2011-2022, in contrast to the decreasing trend in $Chla_s$, $Chla_d$ show

an increasing trend both in model and observations (Figure 5b). Table 4 confirms this increasing trend, showing a similar significant positive slope of the $Chla_d$ post-2011 trend in the model (S=1.31 mg m$^{-2}$ yr$^{-1}$) and observations (S=1.43 mg m$^{-2}$ yr$^{-1}$).

To understand the drivers of the decreasing trend in $Chla_s$ over 2011–2022, we first show the interannual variability of observational mixed layer depth ($z_m$) (Figure 5c) and surface chlorophyll concentration ($Chla_{c,s}$) from the model (Figure

6a). Figure 5c demonstrates a decreasing trend in $z_m$ since 2011 (S=-2.57 m yr$^{-1}$ in Table 4). Similarly, the chlorophyll concentration shows a decreasing trend ($S$=-0.002 mg m$^{-3}$ yr$^{-1}$) which mirrors the decrease in chlorophyll stocks. $\Delta$ of $Chla_{c,s}$ is -26.8 %, which means chlorophyll concentration decreased by approximately 27 % from 2011 to 2022. This further confirms the decreasing trend of $Chla_s$ is related to the decrease in $Chla_{c,s}$ rather than purely driven by a decreasing surface layer water volume.



**Table 4.** The trend and change of state variables from the model over 2011–2022 and the relationship between chlorophyll biomass from model and observation. In this table, $Chla_s$ and $Chla_d$ represent the surface and subsurface integration of the chlorophyll respectively. $Chla_{c,s}$, $z_m$, $\chi_s$, $I_s$, $P_{c,s}$, $N_{c,s}$ and $Z_{c,s}$ represent the deseasonalised MLD, surface chlorophyll concentration, surface C:Chl ratio, surface light, surface phytoplankton concentration, surface DIN concentration and surface zooplankton concentration from the model. Similarly, $Chla_{c,d}$, $I_d$, $P_{c,d}$, $N_{c,d}$, and $Z_{c,d}$ represent the chlorophyll concentration, light, phytoplankton concentration, DIN concentration and zooplankton concentration at the subsurface layer from the model. $S$ means the slope of the post-2011 trend of parameters mentioned above. All $S$ are significant except for the trend of $Z_{c,s}$. The unit of $S$ for each parameter is shown as $[S]$ in square brackets. $\Delta$ denotes the percentage change in the fitted model output for each parameter described above between 2011-01-01 and 2022-12-31. The values from the observations are shown in parenthesis and the rest of values are from model. The correlation coefficient, $Corr.(model, obs)$, shows the relationship between the deseasonalised full timeseries over 1990–2022 in the model and observation; its corresponding p-value is shown in angle brackets. Similarly, $Corr.(model, obs)_{2011}$ shows the relationship between the deseasonalised timeseries over 2011-2022 in the model and observation.

| | $S$ | $[S]$ | $\Delta$ [%] | $Corr.(model, obs)$ | $Corr.(model, obs)_{2011}$ |
|---|---|---|---|---|---|
| $Chla_s$ | -0.42(-0.41) | [mg m$^{-2}$ yr$^{-1}$] | -68.4 | 0.78<p=0.00> | 0.85<p=0.00> |
| $Chla_{c,s}$ | -0.002 | [mg m$^{-3}$ yr$^{-1}$] | -26.8 | | |
| $z_m$ | -2.57 | [m yr$^{-1}$] | -45.3 | | |
| $\chi_s$ | +4.54 | [yr$^{-1}$] | +19.2 | | |
| $I_s$ | +2.71 | [W m$^{-2}$ yr$^{-1}$] | +26.8 | | |
| $P_{c,s}$ | -0.001 | [mmolN m$^{-3}$ yr$^{-1}$] | -7.7 | | |
| $N_{c,s}$ | -0.000 | [mmolN m$^{-3}$ yr$^{-1}$] | -3.8 | | |
| $Z_{c,s}$ | -0.001 | [mmolN m$^{-3}$ yr$^{-1}$] | -5.1 | | |
| $Chla_d$ | +1.31 (+1.43) | [mg m$^{-2}$ yr$^{-1}$] | + 179.2 | 0.44<p=0.00> | 0.75<p=0.00> |
| $Chla_{c,d}$ | +0.004 | [mg m$^{-3}$ yr$^{-1}$] | +172.9 | | |
| $I_d$ | 0.058 | [W m$^{-2}$ yr$^{-1}$] | +45.5 | | |
| $P_{c,d}$ | +0.008 | [mmolN m$^{-3}$ yr$^{-1}$] | +172.9 | | |
| $N_{c,d}$ | -0.040 | [mmolN m$^{-3}$ yr$^{-1}$] | -24.0 | | |
| $Z_{c,d}$ | +0.006 | [mmolN m$^{-3}$ yr$^{-1}$] | +126.4 | | |

Next to understand the drivers of decreasing $Chla_{c,s}$ post 2011, we also show the surface C:Chl ratio ($\chi_s$), light ($I_s$), phytoplankton ($P_{c,s}$), nutrient ($N_{c,s}$) and zooplankton ($Z_{c,s}$) concentrations from the model (Figures 6b-e). $\chi_s$ and $I_s$ shows an increasing trend since 2011 (Figure 6b), reflected by positive slope of 4.54 yr$^{-1}$ and 2.71 W m$^{-2}$ yr$^{-1}$ respecitvley (see Table 4). From 2011 to 2022, $\chi_s$ and $I_s$ increase by around 19.2 % and 26.8 %. In contrast to surface light trend, $P_{c,s}$ shows a weak decreasing trend ($S$=-0.001 mmolN m$^{-3}$ yr$^{-1}$) and a decrease of 7.7%. $N_{c,s}$ shows a near zero decreasing trend and

$Z_{c,s}$ does not show a significant trend ($p$=0.05), which is also reflected by their small change in percentage of -3.8 % and -5.1 % respectively. Our results suggest that the change in surface chlorophyll concentration is primarily driven by the change in surface light and $\chi_s$ rather than the change in phytoplankton growth. The shoaling of mixed layer depth enhances the light in



the surface layer leading to an increasing C:Chl ratio, which further contributes to a decreasing chlorophyll biomass from 2011 to 2022.

To explore the mechanism behind the increasing $Chla_d$ trend post 2011, we also show the deseasonalised chlorophyll concentration ($Chla_{c,d}$) in Figure 7a. Similar to $Chla_d$, $Chla_{c,d}$ shows an increasing trend (S=0.004 mg m$^{-3}$ yr$^{-1}$) and has increased by 172.9 % from 2011 to 2022 (Table 4). Similarly, this increasing trend is also seen in $I_d$, $P_{c,d}$ and $Z_d$, which show a $S$ of 0.058 W m$^{-2}$ yr$^{-1}$, 0.008 mmolN m$^{-3}$ yr$^{-1}$ and 0.006 mmolN m$^{-3}$ yr$^{-1}$ respectively. A large change in the percentage is also seen in $I_d$ (+45.5 %), $P_{c,d}$ (+172.9 %) and $Z_d$ (+126.4 %). Particularly, the change in percentage in $P_{c,d}$ is identical to

$Chla_{c,d}$, owing to the fact the model has a fixed $\chi_d$. These findings suggest that the increase in subsurface chlorophyll biomass is driven by the increase in phytoplankton concentration, which is a result of an increase in light in the subsurface layer due to the shoaling of mixed layer depth and reduction in $Chla_s$. This mechanism is further confirmed by a decreasing trend ($S$=-0.04 mmolN m$^{-3}$ yr$^{-1}$) and a change in percentage of -24 % in $N_{c,d}$, as more nutrient is taken up with subsurface phytoplankton growth.

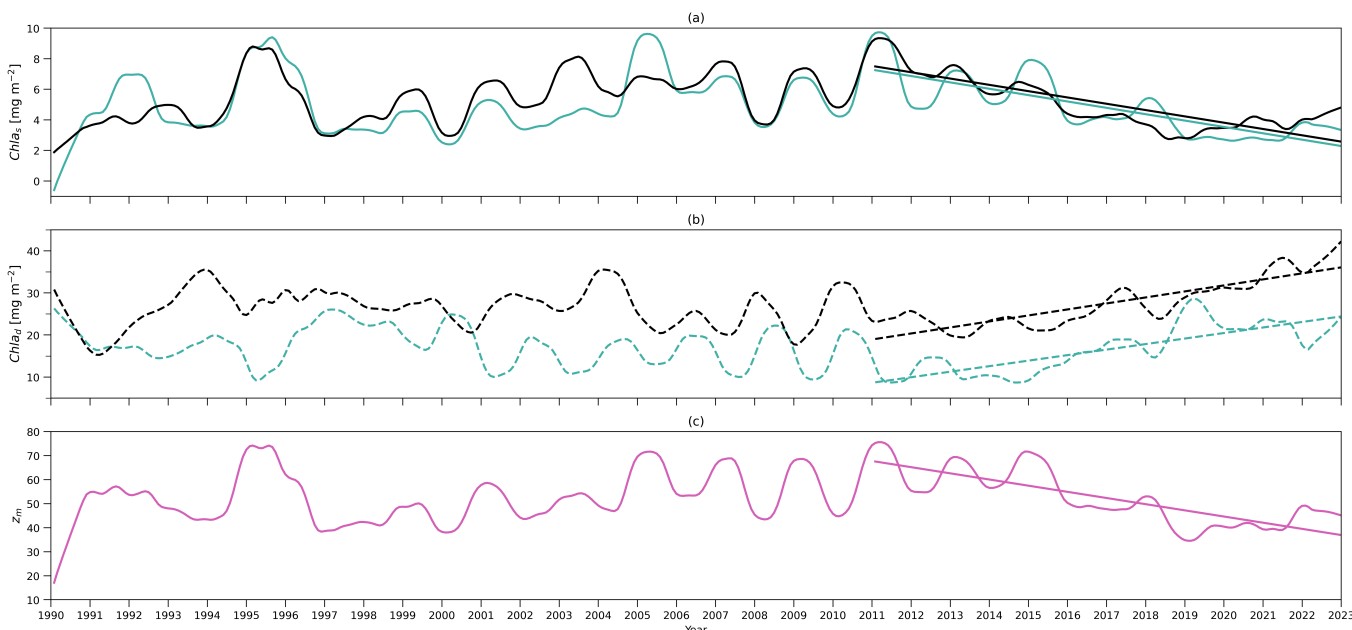

**Figure 5.** (a) deseasonalised chlorophyll integration above $z_m$ (surface layer) timeseries from 1990 to 2022 from the two-layered NPZ model (green) and from observations (black). Straight lines correspond to the linear regressions fitted to deseasonalised data from 2011 to the end of 2022 (post-2011 includes 2011–2022) from model data (green) and observations (black). (b) as in (a) but for deseasonalised chlorophyll-a integration between $z_m$ and $z_{eu}$ (subsurface layer) timeseries. (c) as in (a) but for deseasonalised observational MLD ($z_m$) at BATS.



**Figure 6.** (a) deseasonalised chlorophyll-a concentration vertical-averaged above $z_m$ (surface layer) timeseries from 1990 to 2022 from the two-layered NPZ model. Straight lines correspond to the linear regressions fitted to de-seasonalized data from 2011 to the end of 2022 (post-2011 includes 2011–2022) from model data. (b) deseasonalised surface C:Chl ratio (purple) and light (yellow) timeseries from 1990 to the end of 2022 from the two-layered NPZ model. Purple (yellow) straight dashed line means the linear regressions fitted to de-seasonalized surface C:Chl ratio (light) data from 2011 to the end of 2022 (post-2011 includes 2011–2022) from the two-layered NPZ model. (c) as in (a) but for the deseasonalised surface phytoplankton concentration from the model. (d) as in (e) but for the deseasonalised surface DIN concentration from the model. (e) as in (a) but for the deseasonalised surface zooplankton concentration from the model.



**Figure 7.** (a) deseasonalised chlorophyll-a concentration vertical-averaged between $z_m$ and $z_{eu}$ (subsurface layer) timeseries from 1990 to 2022 from the two-layered NPZ model. Straight lines correspond to the linear regressions fitted to de-seasonalized data from 2011 to the end of 2022 (post-2011 includes 2011–2022) from model data. (b) as in (a) but for deseasonalised subsurface light timeseries from 1990 to the end of 2022 from the two-layered NPZ model. (c) as in (a) but for the deseasonalised subsurface phytoplankton concentration from the model. (d) as in (e) but for the deseasonalised subsurface DIN concentration from the model. (e) as in (a) but for the deseasonalised subsurface zooplankton concentration from the model.



## 4 Discussion


We have built a two-layered NPZ box model of the stratified ocean that assumes the marine ecosystem in the euphotic zone can be partitioned into two distinct communities: one located within the mixed layer and the other situated below the mixed layer and above the very base of the euphotic zone (defined here as the 0.001 % light level). These two communities reside in two different environments: a (typically) nutrient-limited one within the mixed layer and a light-limited one below it.

Model estimates of the time-mean vertically averaged phytoplankton concentration is 0.21 mmolN m$^{-3}$ in the surface layer (above the mixed layer depth) and 0.12 mmolN m$^{-3}$ in the subsurface layer (between the below the mixed layer and above the euphotic zone). These estimates align well with the near-surface value given by Kantha (2004) (0.2 mmolN m$^{-3}$), and the time-mean range of 0.05–0.15 mmolN m$^{-3}$ at between depth of 100–250 m simulated by the coupled physical and NPZD model of Doney et al. (1996). Similarly, this model estimates a time-mean surface chlorophyll concentration of 0.07 mg m$^{-3}$

which is consistent with the estimate (around 0.09 mg m$^{-3}$) modelled by an NPZD model at BATS (Spitz et al., 2001) and the time-mean from observations (around 0.1 mg m$^{-3}$) (Anugerahanti et al., 2020). Also, this model provides a range of 0–0.33 mg m$^{-3}$ for chlorophyll concentration at the subsurface layer, which is consistent with the observational range of 0–0.25 mg m$^{-3}$ and modelling range of 0–0.35 mg m$^{-3}$ given by Anugerahanti et al. (2020).

Regarding zooplankton, limited by fixed observations at 200 m at BATS (Madin et al., 2001), we cannot make a direct com-
parison of our model estimates with these observations. Nonetheless, our model provides comparable time-mean zooplankton concentration estimates of 0.18 mmolN m$^{-3}$ above the mixed layer and 0.1 mmolN m$^{-3}$ below it, which are consistent with those reported by Anugerahanti et al. (2020). They utilize a complex 3D ecosystem model (MEDUSA) to reveal a vertical profile of zooplankton concentration ranging from 0 to 0.3 mmolN m$^{-3}$ at the near-surface decreasing to 0–0.1 mmolN m$^{-3}$ at 200 m (Anugerahanti et al., 2020).

Vertically averaged concentrations of modelled dissolved inorganic nitrogen (DIN) in the two layers align with other studies at BATS. The time-mean of DIN concentration averaged within the mixed layer is around 0.09 mmolN m$^{-3}$ in very good agreement with situ data of DIN at BATS, of around 0.1 mmolN m$^{-3}$ within upper 50 m (Anugerahanti et al., 2020; Hurtt and Armstrong, 1999). The DIN concentration within the mixed-layer shows a range of 0.04–0.5 mmolN m$^{-3}$ which is also consistent with the range (0–0.6 mmolN m$^{-3}$) modelled by Spitz et al. (2001). In addition to the surface layer, the subsurface
time-mean DIN concentration vertically averaged below the mixed layer and above the euphotic zone (1.73 mmolN m$^{-3}$) in our model, agrees with the observational nitrate concentration record (1.75 mmolN m$^{-3}$; Anugerahanti et al., 2018) and modelling output at 160m (of around 2 mmolN m$^{-3}$) from a 1D algal group-based phytoplankton model (Salihoglu et al., 2008).

Our model simulates the vertical-integrated chlorophyll for two ecosystems using a dynamical model of the C:Chl ratio at the
surface layer and a fixed C:Chl ratio at the subsurface layer. The chlorophyll integration within the mixed layer in this model shows a strong seasonal cycle with a maximum (13.63 mg m$^{-2}$) in spring and a minimum (0.52 mg m$^{-2}$) in summer, in good agreement with in situ observations at BATS. Furthermore, this seasonality reconciles closely with the median climatology (around 2.5–18.5 mg m$^{-2}$) of the surface chlorophyll integration partitioned from the vertical profiles using a sigmoid-based





function (Viljoen et al., 2024). Although different integration boundaries make more direct comparisons with other studies
difficult, the range of chlorophyll integrations is also comparable with those modelling and observations reported by Doney
et al. (1996) within the upper 140 m (10–30 mg m$^{-2}$) at BATS.

Both model and in situ data show that phytoplankton in the two layers have a contrasting seasonality at BATS, which is also
seen in statistical partition results of (Viljoen et al., 2024) and in other seasonally stratified regions like the northern Red Sea
(Brewin et al., 2022). The subsurface chlorophyll stock shows a seasonality with a minimum (1.89 mg m$^{-2}$) in spring and a
maximum (39.94 mg m$^{-2}$) in autumn. The maximum matches the in situ measurements (35.24 mg m$^{-2}$), while the minimum
displays a larger discrepancy from observations (11.57 mg m$^{-2}$) in spring. However, this minimum value estimated from
our two-layered model agrees with the minimum (near zero) of the chlorophyll integration below the mixed layer partitioned
using a conceptual model applied to vertical profiles in Viljoen et al. (2024). The former maximum from our models is also
comparable but higher than the maximum of their median chlorophyll integration climatology (around 20 mg m$^{-2}$) (Viljoen
et al., 2024). This discrepancy may be attributed to the deeper integration boundary used in our model and also the different
choice of climatology (time-mean and median) (Viljoen et al., 2024). Given that the measurements below 50 m are scarcer,
especially in spring (Anugerahanti et al., 2020), than near-surface measurements, the values predicted by our model may still
be considered reasonable.

This model simulates a strong seasonality in DIN stocks accumulated within the mixed layer, in agreement with in situ
data (Fig. 4c and g). It shows a peak (11.04 mmolN m$^{-2}$) in spring, agreeing with the peak value in situ measurements
(10.25 mmolN m$^{-2}$) of NO$_2$+NO$_3$ stocks, and a minimum (1.21 mmolN m$^{-2}$) in summer, slightly higher than shown in
observations (0.04 mmolN m$^{-2}$). These lower values (near zero) of in situ measurements during summer and autumn are
limited by the sensitivity of the method used for measuring DIN (Steinberg et al., 2001; Lomas et al., 2013), unable to resolve
low concentrations of DIN. Thus, our estimate could be a reasonable representative of the seasonal minimum surface DIN
stocks.

Our results underscore the challenge of accurately simulating subsurface communities, likely due to difficulties in accurately
determining parameters for the subsurface layer. Modelling subsurface nitrate dynamics, especially seasonality, is particularly
challenging with a simplified model. Despite this, our model aligns with observed time-mean DIN values (around 460 mmolN
m$^{-2}$), closely matching NO$_2$+NO$_3$ integrated stocks (468 mmolN m$^{-2}$). Moreover, while uncertainties in deep-ocean nitrate
dynamics remain, these have minimal effect on subsurface phytoplankton growth which is driven by subsurface light. Sensi-
tivity analysis further emphasizes the importance of determining subsurface parameters for simulating subsurface chlorophyll
and nutrient dynamics. Although the parameters for the subsurface layer in our model are derived from the assimilation of the
observational data at BATS (e.g., Schartau and Oschlies, 2003), there is a need for more precise parameterizations tailored to
the specific subsurface environment to enhance our understanding of phytoplankton communities' responses to climate change.
This can only be achieved through modellers working together with observationalists, targeting key measurements of important
subsurface parameters, and improving understanding of controls on subsurface phytoplankton.

Some studies have observed increasing trends in integrated chlorophyll at BATS (e.g. 150m-integration from 1989 to 2003
Saba et al., 2010). However, recently, Viljoen et al. (2024) observed the opposite trends in surface and subsurface chlorophyll





integration between 2011 to 2022. This opposite trend is also captured in this two-layered model. Our model reveals that shoaling mixed layer depth increases the light availability both at the surface and subsurface layers. However, the two phytoplankton communities at the surface and subsurface layers show different responses to this increase in light intensity. At the surface layer, the C:Chl ratio in this model shows an increasing trend over 2011-2022, which is also seen from Viljoen et al. (2024). This change in C:Chl ratio explains most of the change in surface chlorophyll biomass. Although our model shows a decreasing trend in phytoplankton concentration whereas Viljoen et al. (2024) showed no trend in surface, this change is small (7.7%), and there is little trend in surface nutrient concentration, and no trend in zooplankton. These support the mechanism at the surface layer observed in Viljoen et al. (2024), that intensifying light does not directly drive a decrease in carbon stocks, instead, it changes the physiology of phytoplankton by increasing the C:Chl ratio which leads to a decreasing chlorophyll biomass over 2011-2022. Different from the surface layer, an increasing trend in chlorophyll stocks is still seen from both observations and our model. This is because the growth of phytoplankton is purely driven by the light in the subsurface and the C:Chl ratio is fixed. The intensifying light directly leads to an increase in phytoplankton and chlorophyll biomass, confirmed by the large consistent change in subsurface light, chlorophyll, phytoplankton and zooplankton concentration, which also agrees with the observations in Viljoen et al. (2024). Understanding these mechanisms will help improve projections of future chlorophyll stocks in stratified systems with ocean warming.

Despite our model providing valuable insights into vertical variations in plankton community composition and helping to understand drivers of trends during the past decade at BATS, it is important to acknowledge its limitations. The processes designed in this model do not incorporate all the key biogeochemical processes in stratified systems, such as nitrogen fixation and iron limitation, which should be considered in future developments. Moreover, the model does not explicitly account for important biological components such as bacteria, viruses, and detritus, which play a crucial role in nutrient cycling and ecosystem functioning. Our model does not incorporate the diversity of phytoplankton or zooplankton within the two layers. It also assumes the conservation of phytoplankton and nutrients within the mixed layer during shoaling, which may overestimate the surface phytoplankton, as they could be lost from the surface layer when mixed layer shoals. We also assume that the two vertically separated communities of phytoplankton and zooplankton do not directly interact, only implicitly through the exchange in nutrients between layers. Good agreement between model output and observations supports this assumption. Nevertheless, future work should investigate if an explicit interaction between ecosystems would further improve model simulations. There are also other limitations related to model application. As a two-layered box model, coupling it with other more complex physical models will not be straightforward. Furthermore, the parameters presented in this study are specifically designed for the BATS site and will likely require adjustment for other stratified locations.

## 5 Summary

We developed a two-layer NPZ model for stratified oceans, partitioning the euphotic zone into a surface layer above the mixed layer depth and a subsurface layer below it for the first time. The model applies distinct parameters in each layer to capture the contrasting environmental conditions for phytoplankton growth at BATS. This model also managed to simulate the



chlorophyll seasonal and interannual variability both at the surface and subsurface layers, reproducing observed contrasting trends in chlorophyll between two layers over the past decade. Furthermore, the model identifies distinct mechanisms driving these contrasting trends offering valuable insights for projecting chlorophyll dynamics in stratified oceans. While the model

535    provides meaningful results, we acknowledge its limitations and suggest directions for future work.

*Code and data availability.* BATS data used in this study were acquired freely from the BATS data server (http://bats.bios.edu/bats-data/) for Niskin bottle nutrient data and the BATS project page at the Biological-Chemical Oceanography Data Management Office (https://www.bco-dmo.org/project/2124) for HPLC chlorophyll-a and CTD temperature data (Johnson et al., 2023, 2024). The two-layered model function codes are openly on a GitHub page (https://github.com/Qicodediary/two-layered-ecosystem-model). This page includes examples of Jupyter

540    Notebook Python Scripts, displaying how to run the model and visualize the output.

*Author contributions.* QZ and RJWB designed the model. QZ developed the model code and performed the simulations, with guidance from RJWB. JV assisted in preparing field data for model evaluation and XS made useful recommendations. QZ prepared the manuscript with contributions from all co-authors.

*Competing interests.* The authors declare that they have no conflict of interest.

545    *Acknowledgements.* We sincerely thank all the researchers, technicians, and data managers who have contributed to the BATS site, creating an invaluable wealth of data since sampling started. This work was funded by a UK Research and Innovation Future Leader Fellowship (MR/V022792/1) awarded to Dr. Robert J.W. Brewin. We thank Dr. Shubha Sathyendranath for comprehensive comments, Dr. Zarko Kovac for constructive suggestions for model development, and we thank the MEC team led by Dr. Fanny M. Monteiro for offering invaluable initial insights for ecosystem modelling.



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
