# Peer review of "Simulating vertical phytoplankton dynamics in a stratified ocean using a two-layered ecosystem model"

_EGUsphere, 2024_

## Author Comment (AC1)

**Review comments for the manuscript "Simulating vertical phytoplankton dynamics in a stratified ocean using a two-layered ecosystem model" by Zheng et al.**

Reviewer: Camila Serra Pompei (DTU).

We would like to thank the referee for their thoughtful and constructive comments, which have helped us improve the quality and clarity of our manuscript. Below, we address each comment in detail and outline the revisions made to the text. The referee's comment is shown in black, and our response is beneath in blue. In addition, to improve the manuscript, we have corrected grammatical errors and rewritten the summary section to improve the flow.

In this study, the authors developed a 2-layered NPZ model to better understand the mechanisms driving surface and sub-surface plankton dynamics in a stratified system. The BATS Ocean station is used as a case study, and the model is used to better understand the diverging phytoplankton trends observed in the surface vs subsurface layers in this location. The model provides an interesting avenue to better represent stratified systems with simple NPZ models, where the good model performance compared to field data highlights the strength of the approach. The article is relevant as it shows that different processes drive the dynamics of these two plankton communities, and that changes seen at the surface do not necessarily reflect changes observed in the sub-surface layer.

The manuscript is well written, clear, and model assumptions are well stated. My comments mostly center on the choice of euphotic zone depth and how this affects the representation of the sub-surface community and model comparison with data. There are also some aspects of the sensitivity analysis that need to be clarified.

Thank you for your kind comments. We appreciate the referee's insightful feedback regarding the euphotic zone depth and its influence on the representation of the subsurface community and model comparison. We have carefully addressed these points in the revised manuscript. Below, we provide detailed responses to each comment and outline the corresponding changes made to the text. In the tracked-change manuscript, we highlighted all the changes in blue except for the appendix text.

**Comments regarding the assumption of the euphotic zone depth:**

The assumption of 0.001% might need to be discussed in the discussion section.

We appreciated this suggestion and have added a paragraph to discuss the impact of different assumptions of euphotic zone on subsurface chlorophyll comparisons (see lines 491-497 in page 25 in the tracked-change manuscript).

Although phytoplankton might have been seen growing at these low light levels, is the choice of such a light level representative of where most phytoplankton in the subsurface layer reside?

To clarify this, the choice of the euphotic zone must meet two key criteria in our model. First, the euphotic zone must always extend deeper than the mixed layer depth. This ensures that our simple box model, designed for stratified ocean conditions, consistently maintains two distinct layers, which is essential for mass-conservative. Second, we use the euphotic zone as a reference for setting a lower boundary, indicating that the majority of phytoplankton reside above this depth.

The choice of this low-light level results in the sub-surface layer always having a zeu deeper than 300m, and a depth range (zeu-zm) varying between 150 and 300m. On the other hand, when I look at BATS Chl data, the bulk Chl does not seem to go deeper than 180m, and anything deeper than that has an extremely low Chl concentration.

We show the vertical distribution of chlorophyll measurements in Figure A1 in Appendix A (see page 29). As you said, most of the measurements are above 200 m; however, several measurements extend to around 250 m. We define the euphotic zone to include all available chlorophyll measurements in the upper ocean.

So, it seems to me that the depth range of the sub-surface layer encompasses two layers in the "real world": the deep Chl maximum and whatever is below it.

Yes, it is designed to contain the deep chlorophyll maximum and whatever is living below it, until the bottom boundary we defined.

So, in short, could the authors discuss how this assumption affects the model compared to the real world and how it affects its comparison to field data for the sub-surface layer?

Yes, following your comment, we have done a sensitivity analysis (see appendix A in pages 27-31) and discussed this below.

The above comment specially applies in regard to the method chosen to compare the subsurface layer with field data: first field Chl is averaged between zm and zk and then multiplied by the thickness of the sub-surface layer in the model (zeu-zm). Can the authors mention if the sub-surface layer is representative of the zk-zm depth range?

We assume that the subsurface layer in model (blueline minus redline in Figure A1) is a representative of the $z_k$-$z_m$ depth. Figure A1 shows that, the deepest measurement above the euphotic zone as defined in the model, is relatively close to the modelled euphotic zone depth at each time step, although there is a difference of approximately 50 m. Thus, we believe this assumption is reasonable.

The value of zk is missing in the text and should be provided.

The value of $z_k$ varies at different time steps because the measurements are not uniformly distributed. In the manuscript, we defined $z_k$ as the deepest depth level just shallower than the euphotic zone ($z_{eu}$). Therefore, it is also dependent on the euphotic zone definition. To clarify the values of $z_k$ for readers, we have added Figure A1 in an appendix to illustrating the definition of $z_k$ explicitly and revised the sentences in lines 252-254 in page 11.

If possible, it could have been nice to have some additional figures (e.g. in an appendix) showing the vertical distribution of the field data used. This will help the reader have a better notion of the effect of the data transformation for model comparison.

Thank you for the suggestions. We have added Figure A1 in Appendix A (see page 29) to show the vertical distribution of the field data used.

Does the zeu assumption affect the light experienced in the sub-surface layer?

First, we apologise for the misunderstanding caused by a typo in this manuscript. Our definition of euphotic zone is based on 0.0001% light level, calculated by 13.8/Kd. We have carefully corrected this in the revised manuscript.

The $z_{eu}$ assumption does not significantly affect the subsurface light magnitude (Figure R1 below) because we only use vertically averaged subsurface light in this model. However, there are naturally differences in subsurface light in the model when using different assumptions of $z_{eu}$ in the summer.

We have included a sensitivity analysis of euphotic zone in Appendix A where a detailed description is provided (see pages 27-28). Here, Figure R1 below compares subsurface light from the model using three different definitions of $z_{eu}$, and shows different definitions of $z_{eu}$ have only a small influence on subsurface light, with the largest differences seen during summer.

[Figure]

Figure R1: Subsurface light from the model running with different euphotic zone defined as 0.001% (green line), 0.0001% (black line, definition in manuscript) and 0.00001% (yellow line) light level.

Next, we examine the seasonality of the subsurface light time series (Figure R2). Figure R2 shows that the green line consistently has the highest values, while the yellow line remains the lowest. This indicates that subsurface light intensity is stronger when modelled with a shallower $z_{eu}$ (green line) and weaker when based on a deeper $z_{eu}$ (yellow line). This difference is particularly pronounced in summer (Jun-Aug), but again overall, these differences are relatively small.

[Figure]

Figure R2: Subsurface light averaged seasonality from the model running with different euphotic zone defined as 0.001% (green line), 0.0001% (black line, definition in manuscript) and 0.00001% (yellow line) light level.

How would $Chl_{c,d}$ and $Chl_d$ change when choosing a different $z_{eu}$ value? Given the exponential decay of PAR and its averaging to use as input in the model, I expect $Chl_{c,d}$ not to change much, but $Chl_d$ will certainly change (simply due to its multiplication with $z_{eu}$). How would this affect the data comparison? In other words, it would be good to do a sensitivity analysis on the 0.001% value assumed to define $z_{eu}$.

Following your suggestions, we have conducted a sensitivity analysis and included the key results in Appendix A (see pages 27-31) to thoroughly address the questions raised above. We ran the model with a shallower $z_{eu}$ defined at 0.001% ($z_{eu}$=11.5/Kds) and a deeper 0.00001% ($z_{eu}$=16.1/Kds) surface light level respectively to evaluate how these definitions influence the comparison of modelled chlorophyll with observations. First, we found that different definitions of $z_{eu}$ do not significantly affect subsurface chlorophyll concentrations magnitude ($Chla_{c,d}$) (see Figure R3 below), however, they do influence the seasonal peak values of $Chla_{c,d}$ Thus, we also show the climatology of $Chla_{c,d}$ in Figure R4 below. It shows that, the $Chla_{c,d}$ simulated with a shallower $z_{eu}$ (green) is consistently higher, while weaker when based on a deeper $z_{eu}$ (yellow line). This is consistent with changes in subsurface light illustrated above (Figures R1 and R2). The largest difference in subsurface $Chla_{c,d}$ modelled from different $z_{eu}$ occurs in summer.

[Figure]

Figure R3: Subsurface chlorophyll concentration ($Chla_{c,d}$) from the model running with different euphotic zone defined as 0.001% (green line), 0.0001% (black line, definition in manuscript) and 0.00001% (yellow line) light level.

[Figure]

Figure R4: Subsurface chlorophyll concentration ($Chla_{c,d}$) climatology from the model running with different euphotic zone defined as 0.001% (green line), 0.0001% (black line, definition in manuscript) and 0.00001% (yellow line) light level.

Subsurface chlorophyll integration ($Chl_d$) does indeed vary with changes in the bottom boundary. The changes in euphotic zone in the model also correspondingly changes the observational integration, as the observational integration is calculated by a representative concentration multiplied by model's subsurface layer thickness. However, when comparing the model results with observations, we found different definitions of euphotic zone mainly impact on the seasonality comparison of chlorophyll rather than the long-term trend comparison. Therefore, these different definitions do not substantially alter the main findings (see Figure A2 and A3 in Appendix). A detailed explanation of this has been added to Appendix A of the paper (see pages 27-28).

To sum up: I suggest that (i) an additional paragraph in the discussion section is added addressing the zeu choice and how does this compare to the observed depth range of the subsurface plankton community, (ii) a sensitivity analysis is performed on the 0.001% value, and (iii) some figures showing the vertical distribution of the Chl data are provided in an appendix.

Thank you for summarising your suggestions clearly. In response, an additional paragraph has been included in the discussion section (see lines 491-497). A more detailed sensitivity analysis regarding the euphotic zone is available in Appendix A (see pages 27-31). Additionally, a figure illustrating the vertical distribution of the observational chlorophyll data has also been provided in Appendix A (see page 29).

**Comments regarding the sensitivity analysis:**

I am not sure I understand how ze0 has been varied in the sensitivity analysis. Ze0 is the initial value of the euphotic zone, but zeu is not a fixed parameter, it varies with surface light Kd, Chl etc. So, is this value only different in the first time-step and afterwards just follows the light forcing and kd and amount of Chl?

Yes, this value is only fixed in the first timestep to initiate the model. $z_{e0}$ influences the concentrations of P, Z, N, and chlorophyll in the subsurface layer in the first time step (1990-01-01). Subsequently, this initial impact propagates to euphotic zone calculation from the following day onward as the calculation of the euphotic zone depends on chlorophyll. Given that the calculation of P, Z, N and chlorophyll at each time step will always depends on the euphotic zone, P, Z, N and chlorophyll from the previous time step, the impact of $z_{e0}$ is carried forward and integrated into all subsequent calculations.

The same goes for zm0, since zm is externally forced, does this mean that only the value in the first time step is changed and afterwards all is the same as the MLD forcing?

Yes, this value is only fixed in the first timestep to initiate the model. Similar to $z_{e0}$, $z_{m0}$ will impact the concentration and integration of P, Z, N, and chlorophyll in the subsurface layer at the initial timestep (1990-01-01). The calculation of P, Z, N and chlorophyll in both two layers at subsequent time steps will requires the $z_m$ (MLD forcing), and P, Z, N and chlorophyll from the previous timestep. However, different from the $z_{e0}$'s impact, the $z_m$ is externally forced.

Or does this initial condition somehow affect zm over the entire time-series?

No, it does not affect $z_m$. Because $z_m$ timeseries is an input for this model. We use observational $z_m$ timeseries from BATS site in this study.

L188, why is the initial value of zm set as the mean of the MLD? shouldn't it be set as the initial value of the MLD provided by the data at the starting date of the model? Please provide an explanation on the choice of initial conditions for these forcing variables, why are they not the value provided by the forcing input variables on the corresponding day of the time series?

This model requires a data from the day preceding the first date of the output time series to start the model. For example, if we want to output modelled results from 1990-01-01, we require all initial values on 1989-12-31. However, we do not have $z_{m0}$ on 1989-12-31, unfortunately. We also aim to minimize uncertainty arising from selecting observations on a single specific day. Therefore, we opted to use the time-mean value of MLD as $z_{m0}$.

In terms of $z_{e0}$, it is more challenging to define $z_{e0}$ compared to $z_{m0}$, since no direct observational time series of the euphotic zone depth is available. Instead, the euphotic zone depth is calculated internally within the model, requiring an initial guess. Therefore, the approach used for the MLD (time-mean value) is not applicable. According to Figure 2A in Anugerahanti et al. (2020) (https://doi.org/10.3389/fmars.2020.00549 ), a vertical time-mean profile of chlorophyll at BATS from their 1D modelling output and in-situ data indicates chlorophyll concentration at 250m is nearly zero. Thus, we used 250m as the initial value of euphotic zone to initialise the model.

We also used the approximate time-mean value of each variable (N, P, Z, chlorophyll) from other literatures to initialise the model because we do not have the data on 1989-12-31, and we wanted to minimize uncertainty arising from selecting observations on a single specific day.

**Other comments:**

L 225 Just to clarify, is the "methodology outlined by Viljoen et al. (2024)" the one explained in the rest of this paragraph?

The "methodology outlined by Viljoen et al. (2024)" explains in the rest of this paragraph.

Or is it something different?

No, it is the same.

L118 "but also involves the phytoplankton and zooplankton excursion from the surface layer (Eq. (14))" this makes it sound as if there was a vertical migration process. Perhaps rephrase this sentence as some kind of remineralization of dead plankton and sloppy feeding.

Thanks. The sentence has now been rephrased to: "It not only involves the phytoplankton and zooplankton cycle in the subsurface layer but also the remineralisation of nutrients from some of the dead phytoplankton and zooplankton from the surface layer (Eq. (14))." (see lines 125 in page 5)

L124 mention in this part of the text that table 1 shows a description of the parameters.

We have now added 'with all related parameters described in Table 1' at the end of this sentence (see line 130 in page 6).

L147, please indicate in this part of the text the units of N, P, Z and Chl in the model.

We have now added a sentence to explain the units of N, P, Z and chlorophyll in the model. Please see lines 155-156 in page 7 of the revised manuscript.

Equation 11, remove the multiplication signs (*) for consistency with the rest of equations. The (C:N) ratio could be given a parameter name (e.g. $Q_{C:N}$, or whatever you want to call it), since phytoplankton C:N ratio is fixed in the model. I also suggest changing "MWc" to contain only one capital letter and have the rest of letters as subscripts, as not to be mistaken as two different parameters.

We really appreciate these comments, and have removed the (*) from the rest of equations (see Eq. 11 in page 6). We gave the C:N ratio a parameter name of $Q_{C:N}$ and we also gave MWC a parameter name of $M_c$. Accordingly, we have changed those names in Eq. 11 and 20, in lines 164-165 (in page 7), in Table 1 (in page 9), in line 227 (in page 10), and in Figure 2's caption (in page 12).

Mention earlier in the text that the model has not been spinned-up but that it converges quickly. This is mentioned in the results section, but I was wondering about it while reading the methods.

Thank you for this suggestion. We have added this information in the methodology (lines 245-246 in page 11).

L269, I would not use the word "histogram", as the figure is not a histogram. Perhaps "bar-plot" or simply refer to the figure.

Thank you pointing this out. We have now used bar-plot to refer to the figure (line 281 in page 12).

L283 typo in "notebaly"

Thank you for correcting the typo. We have corrected it to "notably" (line 295 in page 13).

L334 "modelling" should probably be "modelled"

Thank you for your suggestion, we have changed it to modelled (line 346 in page 16)

L335 "is very similar to in observations" should probably be rephrased.

Thank you for these suggestions. This sentence has been rephrased to "The standard deviation from the model (6.94 mg m$^{-2}$) exhibits strong similarity to the observations" (see line 348 in page 17).

---

## Author Comment (AC2)

**Reviewer 2 comments for the manuscript "Simulating vertical phytoplankton dynamics in a stratified ocean using a two-layered ecosystem model" by Zheng et al.**

We would like to thank the reviewer 2 for their thoughtful and constructive comments, which have helped us improve the quality and clarity of our manuscript. Below, we address each comment in detail and outline the revisions made to the text. The referee's comment is shown in black, and our response is beneath in blue. In addition, to improve the manuscript, we have corrected grammatical errors and rewritten the summary section to improve the flow.

The authors developed a novel two-layered NPZ model to simulate the dynamics of surface and subsurface phytoplankton. Using BATS as the research object and adopting reasonable two-layered different parameter settings, they found opposite trends in surface and subsurface phytoplankton after 2011, and analyzed the reasons for this difference. The characteristics of phytoplankton are very consistent with observations in both seasonal changes and long-term trends, further enhancing the credibility of the research results. By establishing and validating this model, the authors provide support for studying the changing characteristics of ecosystems under current conditions. The design and results of this model may also have great applicability and can be further used to predict ecosystem changes under continued warming conditions in the future.

The manuscript is clearly structured and well written. The authors present a compelling study and vividly illustrate their hypotheses and conclusions using well-designed figures and clear explanations. I have some comments and suggestions that I hope will be helpful.

We really appreciate your constructive comments. Your suggestions have helped us to improve our manuscript. Below we carefully address each question you raised and have improved the content accordingly in the revised manuscript. We provide a detailed response to each comment and outline the corresponding changes made to the text. In the tracked-change manuscript, we highlighted all the changes in blue expect for the appendix.

(Text directly from the article is enclosed in "quotation marks".)

**1. Questions**

(1) The two layers of ecosystems are interconnected by dynamic transport of nutrients and light attenuation. The authors used a fixed mixing coefficient (μm) in this study. In equations (5, 17), there is a linear relationship between the nutrients interacting between the two layers and the depth of the mixed layer. But in the real ocean, this relationship is influenced by wind-driven, temperature gradients, and ocean circulation, and is not necessarily linear. I suggest that the authors further explain the effects of the dynamic transport of nutrients in the model on the change of phytoplankton in the two layers in the discussion section. This not only helps to understand the applicability of the two-layered model under real-world conditions, but also provides a reference for future improvements in the model to better simulate actual ocean processes.

Thank you for this valuable suggestion. We agree that explicitly addressing this point will provide a useful reference for future users. In traditional NPZ modelling, it is common to apply a fixed mixing coefficient to represent simply the mixing processes. To improve upon this, we approximated the interaction between the two layers using a linear relationship that includes the dynamic impacts from mixed layer depth (MLD). However, we acknowledge that, in reality, this relationship is neither strictly linear nor fixed. The extent of nutrient transport from the subsurface layer can influence surface phytoplankton growth and the nutrient cycle. Future improvements in the approximation of mixing processes could enhance the model's ability to simulate phytoplankton and nutrient dynamics more realistically.

In response to your comment, and to potentially add some nonlinearity to the exchange of nutrients between layers, we have explored including an entrainment term explicitly, driven by variations in the MLD, to increase the complexity and realism of nutrient exchanges. Initially, our model did not account for interactions between particulate material across the two layers, driven by MLD variability. However, if the MLD oscillates, particulate material (in the form of phytoplankton, as zooplankton are considered

motile) from different layers could transfer between layers, through entrainment. To include this interaction, we first assumed that at each layer, the dominant phytoplankton is the superior competitor, which means intrusion of phytoplankton into the other layer will result in death (and consequent conversion to nutrient). As our model does not explicitly include a detrital variable, the death of phytoplankton adds to the nutrient pools in each layer. This interaction driven by the MLD was explicitly included in the model (see Appendix B), but did not significantly alter our main findings, confirming the robustness of our simpler, original mixing scheme. A more detailed description and corresponding results can be found in Appendix B (see pages 32-35). The blue colour in Appendix B highlighted the modifications we made to the equations in Sec. 2 in the manuscript.

Considering the comment raised, we have emphasised in the discussion that our model assumes a linear relationship between nutrients interacting between the two layers and the depth of the mixed layer, and that in cases where this relationship is not linear, our model may not be appropriate to use. We have also directed the reader to the additional experiments we did in Appendix B. See lines 543-549 on page 26 of the tracked-change manuscript.

(2) The time series of surface Ns observed in Figure 4(c) has many abrupt changes, while the model seems to follow some kind of seasonal cycle. Besides, in Figure 4(d), the standard deviation of the observed results appears to be much larger than the model, although the time-averaged results (Figure 4h) don't show this (You also mentioned this in the article). Why the observation and models are so different on these? What are the potential consequences of this. (Is it related to the daily time resolution?)

The primary reason why the observations and model differ in terms of nutrients, particularly at the subsurface layer, is that simple NPZ models generally struggle to accurately simulate the nutrient cycle, especially in deeper layers. Unlike more complex 3D ecosystem models, this type of simplified model omits many processes necessary for accurately representing the nitrogen cycle, such as nitrogen fixation. However, these processes may not be essential for addressing the main research questions of our study (simulate phytoplankton dynamics in two layers). At the subsurface layer, phytoplankton growth is only limited by light rather than nutrients, which is the main reason we did not employ a more complex nutrient modelling approach. Additionally, as you mentioned, achieving a precise point-to-point match at high temporal resolution (daily) is particularly challenging.

Regarding surface nutrients, the figure shows that nutrient measurements are very sparse, make it theoretically impossible to achieve a day-to-day match. Nevertheless, our model successfully captures the general seasonal patterns and reasonable magnitudes of subsurface nutrients, indicating that our simplified approach remains appropriate for addressing our primary research objectives. However, we acknowledge your concern and have noted this limitation in the discussion section (lines 536-537), emphasising that incorporating more complex processes in the model may be necessary for addressing scientific questions related to nitrogen cycling.

(3) For the model results, the seasonal cycles in Figure 4(e) and Figure 4(g) looks highly correlated. In contrast, the seasonal cycles in Figure 4(f) and Figure 4(h) looks anti-correlated. While, for the subsurface, there seem to be other processes affecting their relationship in winter and spring (Jan-May). Can you explain these further?

Yes, the strong correlation between surface phytoplankton (Figure 4e) and surface nutrients (Figure 4g) in the model occurs primarily because phytoplankton growth at the surface is nutrient limited. In this layer, light is typically abundant, while nutrients are depleted, thus directly controlling phytoplankton productivity. As surface phytoplankton die, they replenish surface nutrients, further reinforcing this close coupling and positive correlation.

In contrast to the surface layer, the modelled seasonality of phytoplankton (Figure 4f) and nutrients in the subsurface (Figure 4h) exhibit an anti-correlation because subsurface phytoplankton growth is limited by light rather than nutrients. In the subsurface layer, nutrients are abundant, but light is depleted, which means phytoplankton growth is not nutrient-limited, but the phytoplankton uptake of nutrients still reduces nutrient concentrations, creating an inverse relationship. Furthermore, subsurface nutrients are influenced not only by local phytoplankton death but also significantly by remineralisation of surface phytoplankton and zooplankton. During winter and spring (January–May), surface phytoplankton blooms further complicate this relationship: blooms at the surface greatly reduce the subsurface light

availability, limiting subsurface phytoplankton growth. However, these same blooms simultaneously enhance nutrient replenishment in the subsurface through increased remineralisation, making the relationship between subsurface phytoplankton and nutrients particularly complex during this period.

(4) Line 397 "This correlation becomes strong over 2011–2022, reflected by an increase in Corr. to 0.75 (Table 4)": The cause of the improved correlation isn't explored. Why does the correlation improve after 2011?

One possible reason for the improved correlation after 2011 is that the impact of ocean warming became more pronounced during the period 2011–2022. Recent research by Viljoen et al. (2024) (see Figure 2 from https://www.nature.com/articles/s41558-024-02136-6) demonstrates that the surface ocean temperature trend intensified notably from 2011 onward compared to earlier decades (1990–2010), which aligns with our findings for subsurface phytoplankton variability. Indeed, a comparison of variance in de-seasonalised subsurface phytoplankton time series reveals a marked shift. The variance of the de-seasonalised subsurface phytoplankton after 2010. For 1990-2010, it is 19.4 $(mg\ m^{-2})^2$ in the model and 14.9 $(mg\ m^{-2})^2$ in the observations, whereas it increased significantly over 2011–2022, reaching 31.2 $(mg\ m^{-2})^2$ in model and 32.7 $(mg\ m^{-2})^2$ in observations. This reduces the discrepancy in variance between model and observations from 4.5 over 1990-2010 to 1.5 $(mg\ m^{-2})^2$ over 2011-2022, highlighting that the model captures the observed increase in interannual variability associated with intensified ocean warming.

**2. Suggestions for improving the content description**

**Abstract**

(1) "reproducing trends post 2011 caused by ocean warming": I suggest going straight to the specific main trends observed at the surface and in the subsurface.

Thank you for your suggestion. We agree that this modification will make abstract more straightforward to readers. This sentence has been revised (see lines 12-15 in page 1).

(2) "simulating the ecosystem in the subsurface layer was more challenging than the ecosystem in the surface mixed-layer": I would suggest briefly mentioning why simulating the subsurface is more challenging, or specifically stating which aspects (e.g. nutrient dynamics) are not accurate.

Thank you for your suggestions and we have now added a brief explanation to this sentence as suggested (see lines 17-18 in page 1).

**Introduction**

In the fourth paragraph of the introduction section "A wide range of ecosystem models are available…": This part seems a bit information-dense, which may make it difficult for readers to grasp the main points. I propose to categorize the models listed and describe the ecological problems that each type can solve.

We really appreciate the suggestions and agree this paragraph needed to be improved for a broad audience to understand. We have revised the paragraph to highlight the main points, providing a general background to ecosystem modelling (see lines 54-59 in pages 2-3). However, we have not included detailed information about categorising the models and describing the ecological problems that each type can solve. We think such a task would require a more comprehensive review appropriate to a review paper, which was not the goal of our paper.

**Results**

Page 18: For this paragraph "To understand the drivers of the decreasing trend in Chla$_s$ over 2011–2022, we first show…", the authors explain that the vertical-average result is also decreasing, indicating that the decreasing trend is not purely caused by the decrease in surface layer water volume. This

explanation is very good. However, at the beginning of this paragraph, authors mentioned "we first show the interannual variability of observational mixed layer depth…", readers would therefore expect this paragraph to describe the relationship between mixed layer depth and chlorophyll, and/or whether the trend of mixed layer depth affects the trend of chlorophyll. It is recommended to adjust/add appropriate content to make the content complete.

Thank you for this suggestion. We agree and have now modified this paragraph to guide the readers through the key findings (lines 417-418 in page 20). However, the relationship between mixed layer depth and chlorophyll stocks at surface layer is quite complex and we cannot explain them within one paragraph. To help readers link back to the previous paragraph to understand the full story, we also added a note in the following paragraph (lines 429-430 in page 20).

**Discussion**

Line 515-517: It was mentioned that "The processes designed in this model do not incorporate all the key biogeochemical processes in stratified systems, such as nitrogen fixation and iron limitation". Could you further explain how the lack of this part will affect this study?

Yes, this model could benefit from including nitrogen fixation processes by improving the agreement between model and observations that you highlighted in your comments Question (2). Diazotrophs can fix atmospheric nitrogen and introduce them into inorganic format in oceans, which could increase the agreement between observational and modelled nitrogen at the surface layer. Given that the BATS site in the Sargasso Sea is characterised by oligotrophic surface waters often limited by nitrogen availability, including nitrogen fixation could relieve nitrogen limitation, thus influencing phytoplankton growth at the surface. Enhanced surface phytoplankton productivity would subsequently affect subsurface nutrient cycling through increased remineralisation, likely improving the agreement between modelled and observed nitrogen in both layers.

The lack of a process describing iron limitation will not significantly affect this study because BATS site is not a high-nutrient, low-chlorophyll (HNLC) zone. Typically, HNLC regions are defined by higher nitrate availability but low phytoplankton biomass due to iron limitation. Although including an iron limitation process will not impact this study, it would likely be essential for simulating phytoplankton dynamics in known HNLC areas, such as the Southern Ocean, North Pacific, or Equatorial Pacific.

**3. Minor error**

Informal expressions like "our ocean" and "less is known about it" should be replaced.

Thank you for pointing this out. This text has now been revised (see lines 2 and 5 in page 1).

Figure 3 caption: "The integration of chlorophyll and nutrients from model and observation, and their relationship" can be revised to: The integration of chlorophyll and nutrients from model and observation, and their relationships.

Thank you for your suggestion. We think you meant Table 3. This text has now been revised (see page 17).

Figure 4 caption: "(d) Daily nitrogen stocks at the surface layer": This should be from subsurface.

We really appreciate you pointing this out. The text has now been corrected (see page 16).

Line 499: "Integration between 2011 to 2022" can be revised to: Integration between 2011 and 2022

Thank you for your suggestion. The text has now been revised (see line 518 in page 25).

---

## Referee Report (RR1)

**Review comments (2nd round) for the manuscript "Simulating vertical phytoplankton dynamics in a stratified ocean using a two-layered ecosystem model" by Zheng et al.**

Reviewer: Camila Serra Pompei (DTU).

I appreciate the efforts of the authors to perform the sensitivity analysis that I requested. The authors have also clarified some of my questions. There is only one minor thing that was not fully addressed. When I requested a figure showing field Chl data, I meant to show the actual Chl concentration (i.e. x-axis time, y-axis depth, z-axis Chl concentration), not only the depths where they were sampled (figure A1 in the new manuscript version). It would be great if the figure could be updated. This is relevant because it gives a notion to the reader about how Chl is distributed across depth, and what the two simulated layers are supposedly modelling. Other than that, the manuscript looks good to me.

---

## Author Response (AR2)

**Simulating vertical phytoplankton dynamics in a stratified ocean using a two-layered ecosystem model**

Qi Zheng [1], Johannes J. Viljoen [1], Xuerong Sun [1], Žarko Kovač [2], Shubha Sathyendranath [3], and Robert J.W. Brewin [1]

[1]Centre for Geography and Environmental Science, Department of Earth and Environmental Sciences, Faculty of Environment, Science and Economy, University of Exeter, Cornwall, UK
[2]Department of Physics, Faculty of Science, University of Split, Rudera Boškovića 33, 21000 Split, Croatia.
[3]National Centre for Earth Observation, Plymouth Marine Laboratory, Plymouth, UK

**Correspondence:** Qi Zheng (q.zheng2@exeter.ac.uk)

[revised manuscript text omitted]
. Furthermore, $N_o$ is set to 0.1 mmolN m$^{-3}$, which is the average value of in situ NO$_3$ observations at BATS from 1998-2007 (Anugerahanti et al., 2020). Observational data for zooplankton at different depths is more challenging to find. To keep consistency, we chose the $Z_o = 0.25$ mmolN m$^{-3}$, based on the experiment that produces phytoplankton and nutrient results closely matching in situ data (PPE in Figure 2 in Anugerahanti et al. (2020)). The initial value of mixed layer depth ($z_{m_o}$) is set to be 52 m, the time-mean value of $z_m$ from 1990 to the end of 2022 at the BATS location (data is described below). Given that the time-mean chlorophyll tends to be near zero at 250 m (Anugerahanti et al., 2020), we define the initial value of the euphotic zone ($z_{e_o}$) as 250 m.

[revised manuscript text omitted]

Concerns may be raised regarding the choice of different bottom boundaries when comparing modelled subsurface integrated chlorophyll against the observations. A sensitivity analysis based on different euphotic zone assumptions (see Appendix A) shows that different assumptions primarily influence the seasonality comparison, particularly in summer when the subsurface layer (water column) in the model is thicker compared to winter. A shallower euphotic zone could cause the model to over-
490   estimate integrated chlorophyll in summer whereas a deeper euphotic zone might make observations not suitable for model validation due to lack of measurements. Despite these potential uncertainties related to the euphotic zone, the interannual variability comparison is hardly influenced by different definitions of euphotic zones.

[revised manuscript text omitted]

Despite our model providing valuable insights into vertical variations in plankton community composition and helping to understand drivers of trends during the past decade at BATS, it is important to acknowledge its limitations. The processes
530 designed in this model do not incorporate all the key biogeochemical processes in stratified systems, such as nitrogen fixation and iron limitation, which should be considered in future developments. Moreover, the model does not explicitly account for important biological components such as bacteria, viruses, and detritus, which play a crucial role in nutrient cycling and ecosystem functioning. Our model does not incorporate the diversity of phytoplankton or zooplankton within the two layers. It also assumes the conservation of phytoplankton and nutrients within the mixed layer during shoaling, which may overestimate
535 the surface phytoplankton, as they could be lost from the surface layer when mixed layer shoals. We also assume that the two vertically separated communities of phytoplankton and zooplankton do not directly interact, only implicitly through the exchange in nutrients between layers. Good agreement between model output and observations supports this assumption. Our model assumes a linear relationship between nutrients interacting between the two layers and the depth of the mixed layer. In cases where this relationship is not linear, our model may not be appropriate to use. We have explored including an explicit
540 entrainment term in our model, to capture some nonlinearity (see Appendix B). This term exchanges phytoplankton between

layers through entrainment (term not included for zooplankton assuming they are motile). The entrained phytoplankton are converted directly to nutrients assuming competitive exclusion (see Appendix B). Including this process had little influence on our results at the BATS location. However, this process may play a more important role in other regions. Future work should investigate if an explicit interaction between zooplankton would further improve model simulations. There are also other limitations related to model application. As a two-layered box model, coupling it with other more complex physical models will not be straightforward. Furthermore, the parameters presented in this study are specifically designed for the BATS site and will likely require adjustment for other stratified locations.

**5 Summary**

We developed a two-layer NPZ model for stratified oceans, partitioning the euphotic zone into a surface layer above the mixed layer depth and a subsurface layer below it for the first time. The model applies distinct parameters in each layer to capture the contrasting environmental conditions for phytoplankton growth at BATS.

This simple model was able to simulate the chlorophyll seasonal and interannual variability both at the surface and subsurface layers, reproducing the observed contrasting trends in chlorophyll between two layers over the past decade.

The model simulations, along with the accompanying sensitivity analyses, reveal that the main mechanisms responsible for the decreasing trend in surface chlorophyll in recent years at the Bermuda time series site are two-fold: (a) increasing trend in light reaching the surface of the ocean at the study site and associated photo-acclimation as indicated by changes in the carbon to chlorophyll ratio in phytoplankton; and (b) changes in stratification and associated decrease in the mixed-layer depth. Interestingly, the model also captures with high fidelity the contrasting seasonal patterns in chlorophyll dynamics in the sub-surface layer, as well as the contrasting increasing trend in the subsurface chlorophyll in recent years. Key to the success of the simulation has been the treatment of the phytoplankton communities in the two layers as distinct communities with distinct bio-optical and physiological traits adapted to the local environment.

The study underscores the importance of following changes in both the layers, to be able to appreciate and predict any current or future changes in marine ecosystems as a whole, in response to changing climate. We see here that the decrease in surface chlorophyll as a result of increasing stratification may be compensated by a corresponding increase in subsurface chlorophyll. Though the detrimental effect of increasing stratification on surface chlorophyll is often discussed in the literature, the beneficial effects on the subsurface phytoplankton communities are often overlooked.

Generalising the model to other regions can only be undertaken cautiously, because many of the simplifications applied to the model were designed to mimic the local conditions at the study site, which includes, for example, a high level of decoupling between the two layers due to stratification, which may not be applicable elsewhere. The strength of the model lies in being able to follow the contrasting dynamics of the surface and subsurface populations of phytoplankton as distinct communities that are simulated using model parameters appropriate for each community.

*Code and data availability.* BATS data used in this study were acquired freely from the BATS data server (http://bats.bios.edu/bats-data/) for Niskin bottle nutrient data and the BATS project page at the Biological-Chemical Oceanography Data Management Office (https://www.bco-dmo.org/project/2124) for HPLC chlorophyll-a and CTD temperature data (Johnson et al., 2023, 2024). The two-layered model function codes are openly on a GitHub page (https://github.com/Qicodediary/two-layered-ecosystem-model). This page includes examples of Jupyter Notebook Python Scripts, displaying how to run the model and visualise the output.

**Appendix A: Sensitivity analysis of euphotic zone**

To explore the impact of different euphotic zone definitions on subsurface chlorophyll integration and its comparison with observations, we conducted a sensitivity analysis using alternative euphotic zone criteria.

In the main manuscript, the euphotic zone is defined as the depth corresponding to 0.0001 % of the surface light level. Accordingly, in this sensitivity analysis, we defined the euphotic zone as 0.001 % (shallower euphotic zone) and 0.00001 % (deeper euphotic zone) of surface light level. By running the model with these two extra euphotic zone assumptions, the subsurface chlorophyll integrations are simulated. Accordingly, the observations are recalculated using these new bottom boundaries following the method described in Sec. 2.4.

Figures A2 and A3 demonstrate the subsurface chlorophyll integration based on 0.001 % and 0.00001 % light level-based euphotic zone respectively (green lines). Accordingly, they also show the new observational subsurface chlorophyll integration (black dots) calculated from 0.001% and 0.00001% euphotic zones separately. Compared with Figure 4f, the main effect of different definitions of the euphotic zone on subsurface chlorophyll integration comparison occurs in summer (Jun-Aug). Given that the mixed layer depth is typically shallow in summer, this results in a thicker subsurface layer. In this case, changes in the bottom boundary have a stronger impact on subsurface light and accordingly the chlorophyll concentration.

Specifically, a shallower euphotic zone tends to cause the model to increase subsurface chlorophyll integration in summer (Figure A2b), whereas a deeper euphotic zone leads to reduce it (Figure A3b). The water column used for integration calculations remains consistent across comparisons, so differences between model simulations and observations arise mainly from variations in modelled and observational representative chlorophyll concentrations under different euphotic zone depths. A shallower euphotic zone generally leads the model to increase chlorophyll concentrations in summer. In contrast, a deeper euphotic zone reduces the subsurface chlorophyll concentration in summer. However, we need to be careful about the observational representative concentration. Given that there are no chlorophyll measurements below 0.0001 % based euphotic zone (see Figure A1), a deeper (0.00001 %) euphotic zone can inflate the observational chlorophyll concentration.

Despite these impacts on seasonality comparisons, different euphotic zones have minimal impact on the post-2011 trend comparison in subsurface chlorophyll integration. Figures A2c and A3c show that the modelled trend remains similar to observations, consistent with the results presented in Figure 5b. In conclusion, varying the euphotic zone assumptions primarily affects seasonal comparisons between the model and observations, but does not change our primary findings regarding post-2011 trends.

[Figure]

**Figure A1.**  Vertical distribution of chlorophyll concentration [mg m$^{-3}$] measurements at BATS from 1990 to 2022. Observational data are plotted as a function of depth (from the sea surface (0 m) to the deep) and year, with colour representing chlorophyll concentration.

[Figure]

**Figure A2.** Comparison of chlorophyll integration between the model and observations using a euphotic zone defined by the 0.001% surface light level. (a) Daily chlorophyll stocks at the subsurface layer from the model (green dashed line) and observations (dark hollow dots) from 1990 to 2022. (b) as in (a) but for the time-mean climatology. (c) deseasonalised subsurface chlorophyll integration timeseries from 1990 to 2022 from the two-layered NPZ model (green) and from observations (black). Straight lines correspond to the linear regressions fitted to deseasonalised data from 2011 to the end of 2022 (post-2011 includes 2011–2022) from model data (green) and observations (black).

[Figure]

**Figure A3.** As in Figure A2, but based on the euphotic zone defined as 0.00001% of the surface light level.

**Appendix B: MLD-driven phytoplankton interactions**

To explore the influence of the mixed layer depth pump on two phytoplankton communities across two layers, we include additional interactions of two phytoplankton communities during mixing processes by modifying equations presented in Sec. 2.1. We implicitly assumed that each phytoplankton community dominates competition in each layer. When these communities interact driven by oscillations of MLD, the superior competitor immediately outcompetes the other. Specifically, when mixed layer depth deepens, those phytoplankton in the subsurface layer carried to the surface layer (simulated by $\zeta^+(t)P_{c,d}$ in Eq. B6) directly die off and directly convert to the surface nutrient pool (Eq. B2) as this model excludes the detrital state variable. We also introduced $\zeta^-(t)$ (see Eq. B1) to help simulate the opposite scenario of a shallowing mixed layer during which phytoplankton from the surface layer ($\zeta^-(t)P_{c,d}$) are advected to the subsurface layer, contributing to the subsurface nutrient pool (see Eq. B4).

To maintain consistency with phytoplankton processes, we similarly apply this mixed layer pump concept for nutrients replacing the traditional asymmetrical MLD effect (by deleting $\zeta^+(t)(N_{c,d}-N_{c,s})$). When mixed layer depth deepens (denoted by $\zeta^+(t)$ in Eq. 1), nutrients from the subsurface layer ($N_{c,d}$) are transported to the surface layer, explicitly represented by $\zeta^+(t)N_{c,d}$ in Eq. B2 and B4. In contrast, when mixed layer depth shallows ($\zeta^-(t)$ in Eq. B1), nutrients from the surface layer are advected to the subsurface layer shown as $\zeta^-(t)N_{c,s}$ in Eq. B2 and B4.

In summary, we introduced a new equation (Eq. B1), and adjusted the surface nutrient (Eq. 5) and phytoplankton (Eq. 6) concentration equations to become Eq. B2 and B3 respectively. Similarly, the subsurface nutrient (Eq. 17) and phytoplankton (Eq. 18) concentration equations become Eq. B4 and B5 respectively. All other equations from Section 2.1 remain unchanged.

$$\zeta^-(t) = min(0, \zeta(t)), \tag{B1}$$

$$\frac{dN_{c,s}}{dt} = -G_s P_{c,s} + \mu_p m_s P_{c,s} + (1 - \gamma_s - \mu_g)\Phi_s Z_{c,s} + \mu_z c_s Z_{c,s}^2 - \frac{\zeta(t)}{z_m}N_{c,s}$$
$$+ \frac{(\mu_m z_m)(N_{c,d} - N_{c,s})}{z_m} + \frac{\zeta^+(t)P_{c,d}}{z_m} + \frac{\zeta^+(t)N_{c,d}}{z_m} + \frac{\zeta^-(t)N_{c,s}}{z_m}, \tag{B2}$$

$$\frac{dP_{c,s}}{dt} = G_s P_{c,s} - m_s P_{c,s} - \Phi_s Z_{c,s} - \frac{\zeta(t)}{z_m}P_{c,s} + \frac{\zeta^-(t)P_{c,s}}{z_m}, \tag{B3}$$

$$\frac{dN_{c,d}}{dt} = -G_d P_{c,d} + m_d P_{c,d} + (1 - \gamma_d)\Phi_d Z_{c,d} + c_d Z_{c,d}^2 - \frac{\eta(t)}{z_D}N_{c,d} + \phi$$
$$- \frac{(\mu_m z_m)(N_{c,d} - N_{c,s})}{z_D} - \frac{\zeta^-(t)P_{c,s}}{z_D} - \frac{\zeta^+(t)N_{c,d}}{z_D} - \frac{\zeta^-(t)N_{c,s}}{z_D}, \tag{B4}$$

$$\frac{dP_{c,d}}{dt} = G_d P_{c,d} - m_d P_{c,d} - \Phi_d Z_{c,d} - \frac{\eta(t)}{z_D}P_{c,d} - \frac{\zeta^+(t)P_{c,d}}{z_D}, \tag{B5}$$

Next, running the model with these modified equations, we show the chlorophyll and nutrient integration from the model and observations (Figure B1). The results obtained from these simulations were very similar to those presented in Figure 4. We also present the deseasonalised chlorophyll integration from the model and observations in Figure B2, which also closely

match the patterns in Figure 5. Consequently, our main conclusions remain robust, even when incorporating this more complex mixing scenario driven by mixed layer depth variations.

[Figure]

**Figure B1.** (a) Daily chlorophyll stocks at the surface layer from the model (green solid line) and observations (dark solid dots) from 1990 to 2022. (b) Daily chlorophyll stocks at the subsurface layer from the model (green dashed line) and observations (dark hollow dots) from 1990 to 2022. (c) Daily nitrogen stocks at the surface layer from the model (blue solid line) and observations (dark hollow dots) from 1990 to 2022. (d) Daily nitrogen stocks at the subsurface layer from the model (blue dashed line) and observations (dark hollow dots) from 1990 to 2022. (e-h) as in (a-d) but for their time-mean climatology. Results are from simulations using Eq. B1–B5.

[Figure]

**Figure B2.** (a) Deseasonalised chlorophyll integration above $z_m$ (surface layer) timeseries from 1990 to 2022 from the two-layered NPZ model (green) and from observations (black). Straight lines correspond to the linear regressions fitted to deseasonalised data from 2011 to the end of 2022 (post-2011 includes 2011–2022) from model data (green) and observations (black). (b) as in (a) but for deseasonalised chlorophyll-a integration between $z_m$ and $z_{eu}$ (subsurface layer) timeseries. Results are from simulations using Equations B1-B5.

*Author contributions.* TEXT

QZ and RJWB designed the model. QZ developed the model code and performed the simulations, with guidance from RJWB. JV assisted in preparing field data for model evaluation and XS made useful recommendations. ZK and SS provided key suggestions for improving the manuscript. QZ prepared the manuscript with contributions from all co-authors.

*Competing interests.* The authors declare that they have no conflict of interest.

*Acknowledgements.* We sincerely thank all the researchers, technicians, and data managers who have contributed to the BATS site, creating an invaluable wealth of data since sampling started. This work was funded by a UK Research and Innovation Future Leader Fellowship (MR/V022792/1) awarded to Dr. Robert J.W. Brewin and partly supported by the Croatian Science Foundation under the project number IP-2022-10-8859. This work was also supported by the National Centre for Earth Observation, the Simons Collaboration on Computational Biogeochemical Modeling of Marine Ecosystems (CBIOMES, 549947 to SS), and the European Space Agency's project Tipping points and abrupt changes In the Marine Ecosystem (TIME). 
[revised manuscript text omitted]